# Pegmatite lithium deposits formed within low-temperature country rocks

Jinsheng Zhou [1,2] ✉, Qiang Wang [1,2,3] ✉, He Wang[1,2], Jinlong Ma[1,2], Guanhong Zhu[1,2] & Le Zhang [1,2]

The global climate crisis is likely to lead to a potential supply risk of lithium (Li) over the coming decades. More than half of the world's production of Li is derived from Li-bearing pegmatites. Although pegmatites are widespread, only a small fraction host economically relevant Li mineralization. Revealing which factors cause some pegmatites to be enriched in Li and others to be barren is critical for understanding Li pegmatite formation and for guiding exploration for new Li resources. In this study, we used an approach involving the analysis of natural samples from the Jiajika pegmatite deposit (China), combined with thermal and diffusion modeling. Here we show that Li contents in pegmatites are controlled not only by the initial Li contents in pegmatite melts but also by the temperature of the surrounding country rocks at the time of pegmatite emplacement. Lithium-mineralized pegmatites form preferentially when Li-rich pegmatite melts intrude low-temperature country rocks.

To tackle the global climate crisis, an increasing number of countries have committed to attaining net-zero $CO_2$ emissions by 2050 or soon after[1]. In many developed countries, internal combustion engine vehicles are the greatest contributors to $CO_2$ emissions[2]. Thus, petrol-powered vehicles should be replaced by electric vehicles on a large scale to meet global targets for reducing carbon emissions[1]. Accordingly, many governments (such as those of the United Kingdom, Germany, Canada, the Netherlands, France, and Ireland) and major carmakers (such as General Motors [U.S.], Audi [Germany], and BYD [China]) have announced that they will stop selling petrol-powered vehicles before ~2035[3,4]. Lithium-ion batteries are the key component in electric vehicles, meaning that the demand for lithium (Li) is increasing dramatically. According to predictions of the International Energy Agency, the demand for Li for batteries will grow 30-fold by 2030 and more than 100-fold by 2050 compared with 2020[1]. Although the recycling of Li from batteries has been proposed as a potential solution, such recycling is scarce (<1%) because of technical constraints and limited collection arrangements[5]. Therefore, a potential supply risk for Li is likely to occur in the next few decades. More than half of the world's production of Li is now derived from Li-bearing

minerals (such as spodumene, petalite, and lepidolite) hosted in pegmatite deposits in Australia[6]. The richest Li-bearing ores in nature occur in pegmatites[7], but not all pegmatites have high Li contents. For example, of the ~24,000 pegmatites in the Black Hills district, South Dakota, only 14 mines contain economic quantities of Li[8]. It poses an important question as to the factors that cause some pegmatites to be enriched in Li whereas others are barren.

Although Li is a metallic element, its atomic mass is very small (6.941), only higher than H (1.008) and He (4.003). Together with its low ionic charge and size, Li has very high mobility. Therefore, Li can easily diffuse during high-temperature geological processes[9]. Diffusion process can be well recorded by Li isotopes. Lithium has two stable isotopes (i.e., $^6Li$ and $^7Li$), and $^6Li$ has a higher diffusion rate than $^7Li$ (~3% faster)[10,11]. As a result, diffusion will lead to unique Li isotope profiles. Lithium isotope diffusion profiles have been found around lithium pegmatites as well as granitic plutons[12–14], confirming that lithium diffusion occurs during the emplacement of pegmatites. However, the purpose of these studies was to reveal the kinetic fractionation behavior of lithium isotopes and did not pay attention to the effect of diffusion on Li content as well as Li mineralization of

[1]State Key Laboratory of Isotope Geochemistry, Guangzhou Institute of Geochemistry, Chinese Academy of Sciences, Guangzhou, China. [2]CAS Center for Excellence in Deep Earth Science, Guangzhou, China. [3]College of Earth and Planetary Sciences, University of Chinese Academy of Sciences, Beijing, China. ✉e-mail: jinshengzhou@gig.ac.cn; wqiang@gig.ac.cn

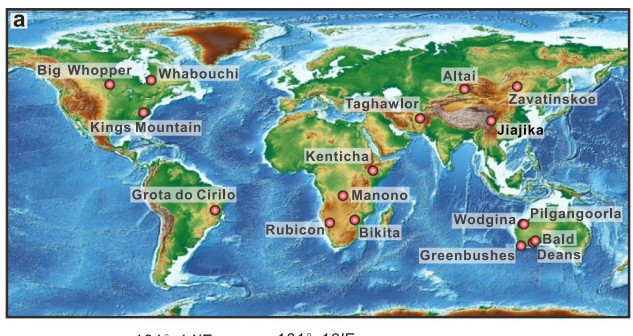

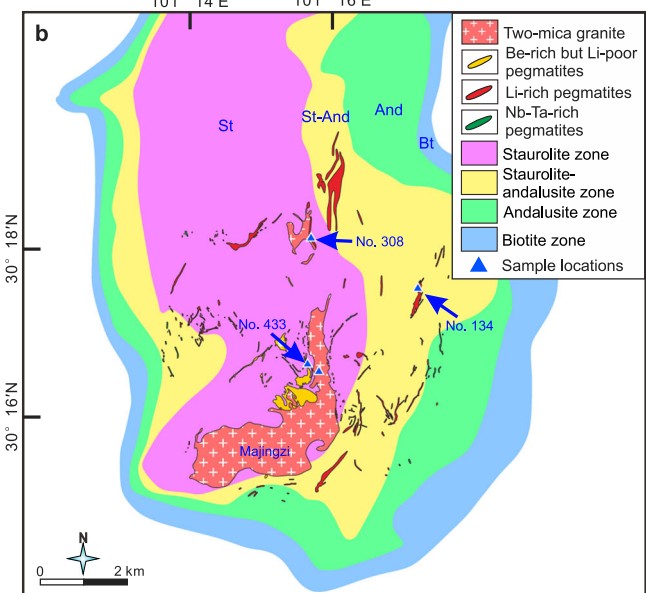

**Fig. 1 | Global distribution of representative pegmatite lithium deposits and geological map of the Jiajika pegmatite deposit. a** Global distribution of representative pegmatite lithium deposits. Basemap was generated using the Generic Mapping Tool (https://www.generic-mapping-tools.org/). **b** Geological map of the Jiajika pegmatite deposit showing sampling locations (blue triangles). The boundaries of pegmatite dikes and granites are from Huang et al.[21]. The boundaries of the metamorphic zones around the Majingzi pluton are from Tang et al.[24] and Fu et al.[27]. From the inner to outer aureoles, they are staurolite, staurolite-andalusite, andalusite, and biotite zones, respectively.

pegmatites[12,13]. Moreover, pegmatites were assumed to be an infinite element Li reservoir in these studies[12,13], meaning that diffusion has no effect on Li content as well as the mineralization ability of pegmatite itself. Based on an observation of substantially higher $\delta^7$Li values of Li-poor pegmatites than Li-rich dikes, a recent study attributed the higher $\delta^7$Li values of Li-poor pegmatites to diffusion-driven fractionation with a long duration, and speculated that diffusion may decrease the Li content of pegmatites[15]. Such a process of Li loss is critical for the formation of pegmatite Li deposits because it can determine the final grade and tonnage of pegmatite ore bodies. However, what factors account for the degree of Li loss by diffusion during pegmatite emplacement remains unclear.

In this study, we utilized an approach involving the analysis of trace elements and Li isotopes of profiles around Li-rich and Li-poor pegmatites in the Jiajika deposit, China, which is a world-class pegmatite Li deposit, combined with comprehensive thermal and diffusion modeling. We find that Li contents in pegmatite dikes are not only dependent on the Li content of pegmatite melts but also on the temperature of country rocks at the time of pegmatite emplacement, with Li-rich pegmatites forming preferentially when Li-rich pegmatite melts intrude low-temperature country rocks.

## Results

### Jiajika pegmatite deposit

World-class Li-pegmatite deposits are found in the eastern Tibetan Plateau region[16], including the Jiajika, Ke'eryin, and Zhawulong deposits[17–19]. Recently, new prospecting targets have been reported from this region[20]. More than 1000 dikes have been discovered in the Jiajika pegmatite deposit, and 509 of these each have an area of >20 m[21]. Of these 509 dikes, ~30 pegmatite bodies contain high abundances of Li[22]. The latest resource estimate for Li in the Jiajika pegmatite deposit is 1.3 million tonnes[22], which exceeds the resource of the Greenbushes pegmatite (0.9 Mt)[16], probably making Jiajika the largest pegmatite Li deposit in the world.

Pegmatite dikes in the Jiajika deposit are distributed around two granitic plutons, the Majingzi and Jiajijiami two-mica granites (Fig. 1b). The exposed area of the former pluton is 5.3 km², and that of the latter is 0.0014 km². Abundant geochronological data for the Jiajika granites and pegmatites have been reported[23], with zircon U–Pb ages clustering in the range of ~220 to ~210 Ma, suggesting a close genetic relationship between the granites and pegmatites. Country rocks in the Jiajika deposit comprise schists and metamorphic sandstones. Five pegmatite rock-type zones can be identified with increasing distance from the Majingzi two-mica granite; i.e., the microcline, microcline–albite, albite, albite–spodumene, and albite–lepidolite pegmatite zones[24]. The Li-rich pegmatite bodies are located mainly within the albite–spodumene pegmatite zone, distal to the Majingzi granite, and the Li-poor but Be-rich pegmatite dikes are proximal to the granite pluton (Fig. 1b), consistent with the classic zoning pattern of metals in pegmatite deposits[7,8,25,26]. Metamorphic zones developed around the Majingzi pluton[24,27]. From the inner to outer aureoles, they are staurolite, staurolite-andalusite, andalusite, and biotite zones, respectively (Fig. 1b), which were formed by the superimposition of a Buchan-type metamorphism upon a Barrovian-type metamorphism[28].

Many pegmatite dikes in the Jiajika deposit have been numbered. For this study, we collected samples from three representative pegmatite dikes (Nos. 134, 308, and 433; Fig. 1b), the Majingzi granite, the surrounding country rocks, and the far-field country rocks. The No. 134 pegmatite is an extremely Li-rich dike and contains 0.28 million tonnes of Li, with an average $Li_2O$ content of 1.38 wt.%[21]. Within the No. 134 pegmatite body, textures and minerals are variable, ranging from coarse to fine-grained textures. The main minerals include spodumene, quartz, muscovite, feldspar, tourmaline, etc. At a location where there is a clear contact boundary between the pegmatite and the country rock, profile samples of the country rock were collected in the horizontal direction perpendicular to the contact boundary (Table 1). The No. 308 pegmatite is also Li bearing but has a lower Li resource, with 0.0028 million tonnes of Li at an average grade of 0.5 wt.%[29]. The internal texture and mineral assemblage of the No. 308 pegmatite are also variable. The main minerals are quartz, spodumene, muscovite, feldspar, tourmaline, etc. Similarly, a country rock profile was selected for sampling (Table 1). The No. 433 dike is a Li-poor but Be-bearing pegmatite, and it is mainly composed of quartz, feldspar, muscovite, tourmaline, etc. The country rock surrounding the No. 433 dike was poorly exposed, and no suitable profile was found for sampling. Only two country rock samples were collected near the dike (<2 m).

### Lithium isotopes and contents

Lithium elemental and isotopic compositions of the Li-rich pegmatites (i.e., the No. 134 pegmatite), Li-bearing pegmatites (i.e., the No. 308 pegmatite), Li-poor pegmatites (i.e., the No. 433 pegmatite), the Majingzi two-mica granites, the country rocks adjacent to the three pegmatites, and the far-field country rocks are listed in Table 1 and plotted in Fig. 2. The Li contents of the five pegmatite samples in the No. 134 pegmatite range from 6051 to 13720 ppm, but the changes in $\delta^7$Li are small (−1.3‰ to +0.1‰). In the country rock profile around the No. 134 pegmatite, the three adjacent samples have approximately

**Table 1 | Li isotope compositions for rocks in the Jiajika pegmatite deposit**

| Pegmatite | Sample | Lithology | Distance from pegmatite contact (m) | Rb (ppm) | Li (ppm) | $\delta^7Li$ (‰) |
|---|---|---|---|---|---|---|
| No. 134 pegmatite | 19JJK01-1 | Pegmatite | | 633 | 12605 | −0.3 |
| | 19JJK01-2 | Pegmatite | | 438 | 13720 | 0.1 |
| | 19JJK01-3 | Pegmatite | | 1125 | 6051 | −1.3 |
| | 19JJK01-4 | Pegmatite | | 789 | 9369 | −0.4 |
| | 19JJK01-5 | Pegmatite | | 913 | 6294 | −0.8 |
| | 19JJK01-6 | Schist | 0.1 | 126 | 233 | 1.7 |
| | 19JJK01-7 | Schist | 1 | 161 | 259 | 0.1 |
| | 19JJK01-8 | Schist | 2 | 143 | 256 | 1.2 |
| | 19JJK01-9 | Schist | 3 | 247 | 1041 | 1.8 |
| | 19JJK01-10 | Schist | 4 | 229 | 1040 | 1.8 |
| | 19JJK01-11 | Schist | 5 | 463 | 1371 | 2.0 |
| | 19JJK01-12 | Schist | 6 | 907 | 1796 | 0.5 |
| | 19JJK01-13 | Schist | 8 | 1173 | 2276 | 1.2 |
| No. 308 pegmatite | 19JJK02-1 | Pegmatite | | 2225 | 7133 | −0.7 |
| | 19JJK02-3 | Pegmatite | | 2863 | 1944 | −0.9 |
| | 19JJK02-4 | Pegmatite | | 1077 | 19385 | −0.8 |
| | 19JJK02-5 | Pegmatite | | 2343 | 1327 | 1.7 |
| | 19JJK02-6-1 | Pegmatite | Near the contact | 1227 | 455 | 1.4 |
| | 19JJK02-6-2 | Schist | 0.1 | 410 | 717 | 1.5 |
| | 19JJK02-7 | Schist | 5 | 177 | 974 | −2.8 |
| | 19JJK02-8 | Schist | 9 | 176 | 389 | −6.7 |
| | 19JJK02-9 | Schist | 13 | 162 | 249 | −11.7 |
| | 19JJK02-10 | Schist | 18 | 150 | 263 | −13.2 |
| | 19JJK02-11 | Schist | 22 | 161 | 239 | −13.8 |
| No. 433 pegmatite | 19JJK05-1 | Pegmatite | | 426 | 92.3 | 5.4 |
| | 19JJK05-2 | Pegmatite | | 1740 | 25.0 | 7.6 |
| | 19JJK05-4 | Schist | <2 | 156 | 894 | 2.5 |
| | 19JJK05-5 | Schist | <2 | 152 | 1049 | 0.9 |
| Majingzi granite | 19JJK03-1 | Two-mica granite | | 292 | 290 | −1.3 |
| | 19JJK03-2 | Two-mica granite | | 296 | 294 | −1.7 |
| | 19JJK04-1 | Two-mica granite | | 304 | 269 | −1.0 |
| | 19JJK04-2 | Two-mica granite | | 301 | 199 | −1.0 |
| Far-field country rock | 19JJK09-1 | Schist | | 168 | 128 | 1.5 |
| | 19JJK09-3 | Schist | | 217 | 92.8 | −1.9 |
| | 19JJK10-2 | Schist | | 31.8 | 14.1 | 1.1 |

identical Li contents and $\delta^7Li$ values (Fig. 3a, c). Starting from the -3-meters position, the Li content gradually increases outwards (Fig. 3a), while the $\delta^7Li$ generally remains unchanged (+0.5‰ to +2.0‰; Fig. 3c). The Li contents of the No. 308 pegmatite fluctuate greatly (455 ppm to 19385 ppm), while the $\delta^7Li$ values are relatively uniform (−0.9‰ to +1.7‰). In the country-rock profile around the No. 308 pegmatite, the Li content gradually decreases (Fig. 3b), and the $\delta^7Li$ also gradually decreases simultaneously (Fig. 3d). An abnormal circumstance occurred in the No. 433 pegmatite: the pegmatite itself has very low Li contents (25 ppm to 92.3 ppm) but very high $\delta^7Li$ values (+5.4‰ to +7.6‰). In contrast to the pegmatite, its adjacent country rocks have higher Li contents (894 to 1049 ppm). The Majingzi granites have relatively identical Li contents (199 to 294 ppm) and $\delta^7Li$ values (−1.7‰ to −1.0‰; Fig. 2). The far-field country rocks (about several kilometers away from the Jiajika deposit) have lower Li contents (14.1 ppm to 128 ppm), and similar $\delta^7Li$ values (−1.9‰ to +1.5‰). We also present the literature data of the Jiajika deposit (Fig. 2), and the distribution patterns of Li contents and Li isotopes are generally consistent with this study: (1) Li-rich pegmatites generally have lower $\delta^7Li$ values than Li-poor pegmatites; (2) Some Li-poor pegmatites have very high $\delta^7Li$ values; (3) The Li contents and $\delta^7Li$ values of the country rocks in the

deposit range greatly, and some samples have abnormally low $\delta^7Li$ values; (4) The Li content and $\delta^7Li$ of the Majingzi granites are relatively homogeneous.

## Discussion

### Migration of Li from pegmatite into country rocks by diffusion

Lithium migration from pegmatites into country (host) rocks has long been recognized[30,31]. The medium for such lithium migration was considered to be fluids and associated induced metasomatism[31]. The country rocks of the three studied pegmatite dikes in Jiajika have differing Li contents and Li isotope profiles (Fig. 3). Country rocks adjacent to the No. 134 pegmatite that has the highest Li abundance of the studied pegmatites, have low and consistent Li contents (ranging from 233 to 259 ppm; the left three data points in Fig. 3a) compared with those adjacent to the No. 308 pegmatite (Fig. 3b). Lithium isotopic compositions of the country rocks adjacent to the No. 134 pegmatite are homogeneous ($\delta^7Li$ = +0.1‰ to +1.7‰; Fig. 3c). In contrast, the country rocks of the No. 308 pegmatite, which has low Li reserves, have relatively high and variable Li contents (239–974 ppm Li; Fig. 3b). More importantly, both Li contents (decreasing from 974 to 239 ppm) and Li isotopic compositions ($\delta^7Li$ decreasing from +1.5‰ to −13.8‰)

show a clear trend of change with increasing distance from the pegmatite–country-rock contact (Fig. 3b, d).

A decreasing trend of Li content with increasing distance from the contact can be ascribed to either fluid metasomatism[31] or the migration of Li by diffusion[12,13,32,33]. These processes can be distinguished by comparing the profiles of Rb and Li in country rocks, as the transfer

distance of Rb from pegmatite into country rocks is close to the infiltration distance of pegmatite-derived fluids[31]. In the country-rock profile line of the No. 308 pegmatite, the gradient profile of Rb content is no more than 5 m long (Fig. 3b; Table 1), whereas the gradient profile of Li content is more than 20 m long (Fig. 3b, d). These observations indicate that the gradient profile of Li content around the pegmatite dike was caused by Li diffusion and that the infiltration of pegmatite-derived fluids works over only very short distances. It is also supported by the Li isotope profile (Fig. 3d), i.e., continuous decreases in Li content and the large fractionation of Li isotope confirm that it is a diffusion-induced profile[12]. Based on the Li diffusion modeling for the No. 308 pegmatite, the measured Li contents and Li isotope profiles are consistent with the simulated profile at 20 years after emplacement (Fig. 4). The cooling time of pegmatite dikes obtained by different methods is variable, ranging from several days[8,34,35] to millenniums[36]. Our results show that the cooling time of the No. 308 pegmatite in Jiajika is ~20 years. These different cooling times may all be reasonable within a pegmatite deposit. Because the conditions and intrusion environments of each dike are different, and the cooling rates vary greatly. Interestingly, No. 433 pegmatite has very low Li contents (25.0–92.3 ppm Li; Table 1), but its adjacent country rocks have very high Li contents (894–1049 ppm Li; Table 1). There may be multiple explanations for this pattern. For example, the No. 433 dike is originally Li-poor and has a very high $\delta^7Li$ value. Its country rocks may have originally had a high Li content, or they were contaminated by other Li-rich pegmatites through diffusion.

## Timescale of pegmatite cooling

Obviously, the amount of Li migration from the pegmatite dike into country rocks depends on the duration of the cooling of pegmatites. The cooling time of pegmatite dike can be constrained by thermal modeling[8,36]. Here, a conductive cooling model (HEAT3D[37]) was employed to track the thermal history of pegmatite dikes after their

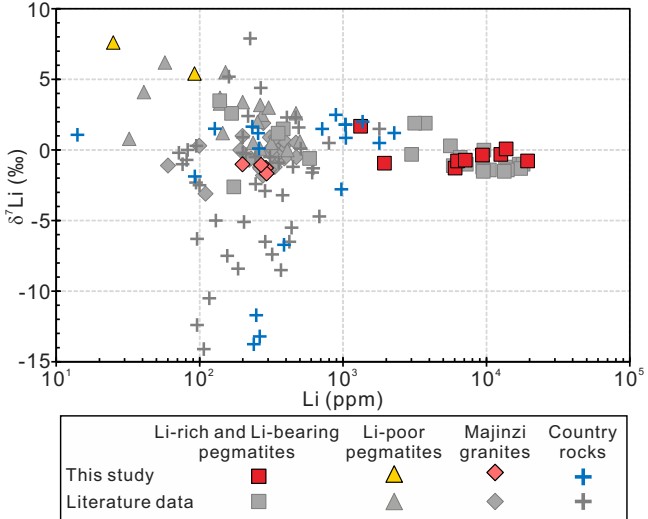

**Fig. 2 | Lithium content vs. $\delta^7Li$ value for different rock types of the Jiajika pegmatite deposit, including the Li-rich pegmatites, Li-poor pegmatites, Majingzi granites, and country rocks.** Data sources include this study (Table 1), Zhao et al.[66], Zhang et al.[67], and Zhang et al.[68]. The error bars are smaller than the symbol size because the standard solution of Li isotope analyses is approximately ±0.52‰ (2 SD).

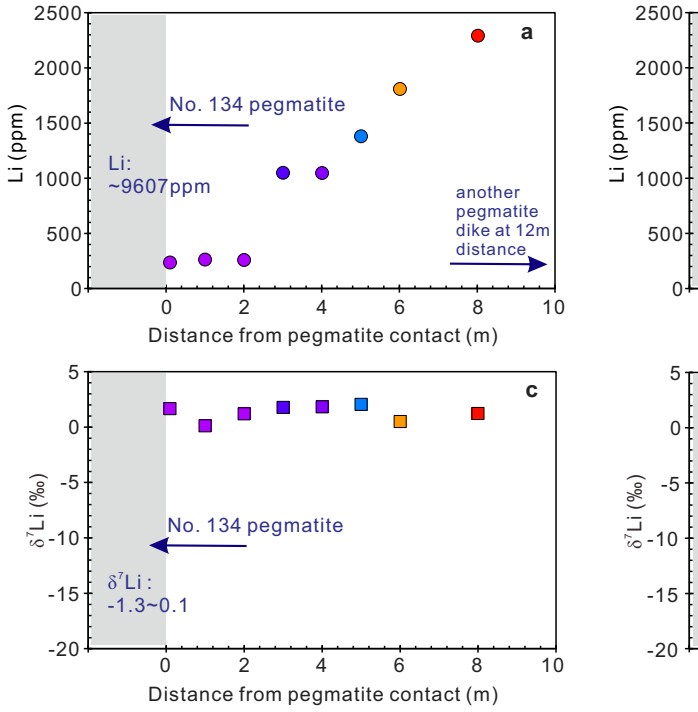

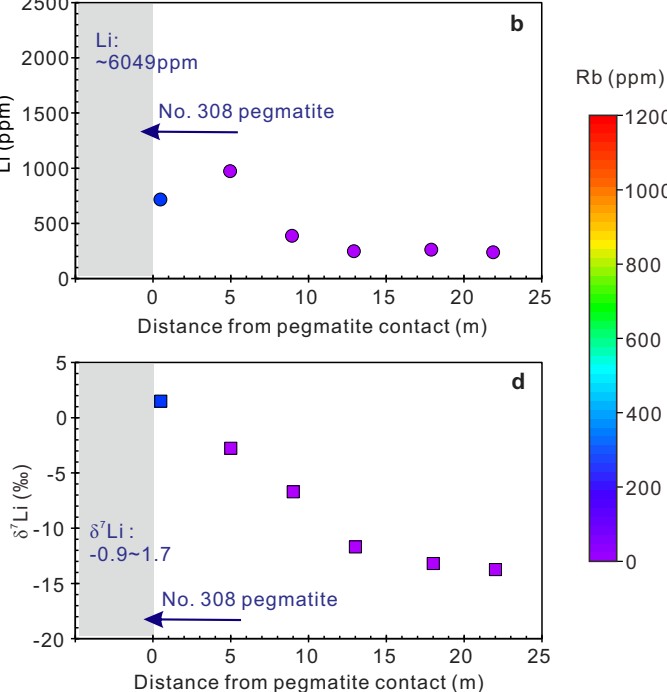

**Fig. 3 | Measured lithium contents and isotopic compositions along profile lines extending outward from the No. 134 and No. 308 pegmatites for the Jiajika pegmatite deposit.** **a** Lithium content profile around the No. 134 pegmatite. We assume that only the three data points on the left are affected by the No. 134 pegmatite, while the gradually increasing Li content on the right is affected by

another pegmatite dike (observed ~12 m distance in the field). This is supported by the changes in Rb content. **b** Lithium content profile around the No. 308 pegmatite. **c** $\delta^7Li$ profile around the No. 134 pegmatite. **b** $\delta^7Li$ profile around the No. 308 pegmatite. The error bars are within symbol size as the standard solution of Li isotope analyses is ~±0.52‰ (2 SD).

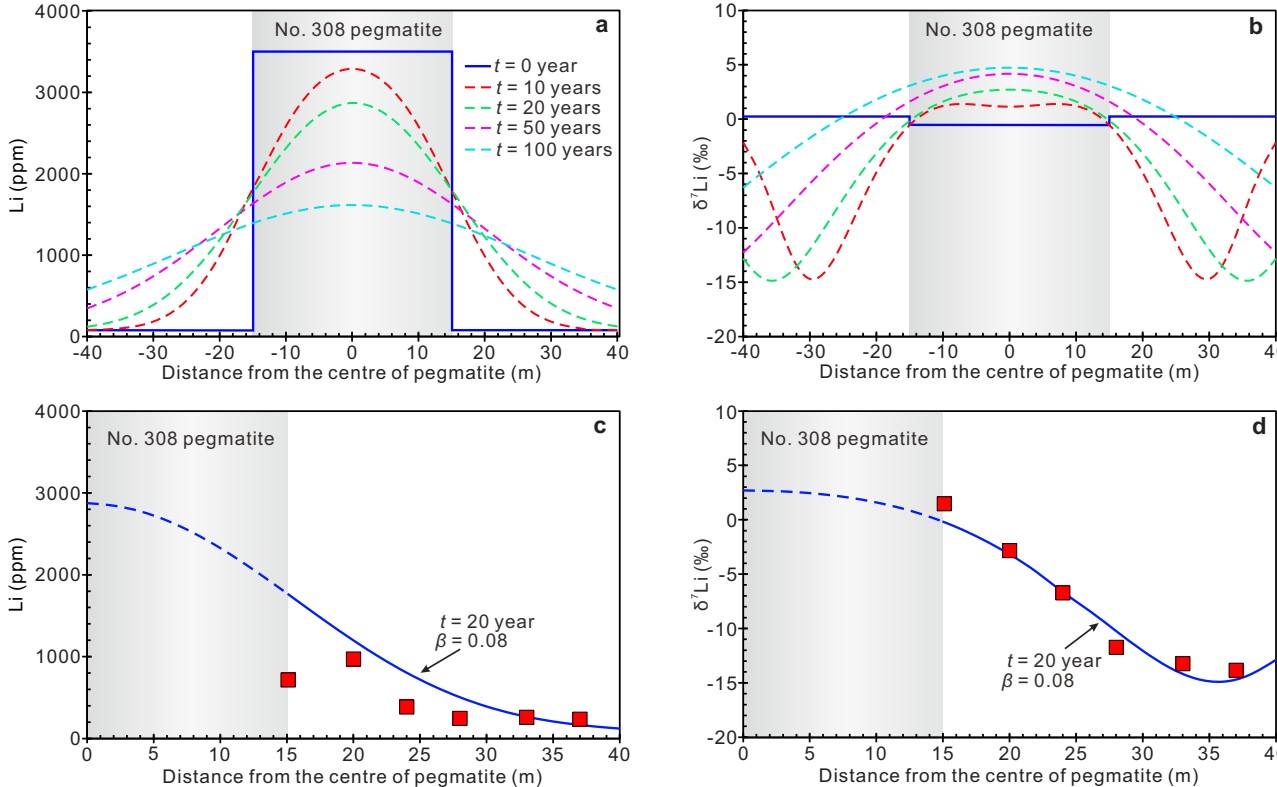

**Fig. 4 | Modeled lithium contents and isotopic compositions along the profile line extending outward from the No. 308 pegmatite. a** Diffusion profiles of Li contents for the No. 308 pegmatite after durations of 10, 20, 50, and 100 years after emplacement, with an effective diffusion coefficient of $10^{-7}$ m$^2$ s$^{-1}$ [12]. **b** Diffusion profiles of $\delta^7$Li values for the No. 308 pegmatite after durations of 10, 20, 50, and 100 years after emplacement. **c** Modeled and measured Li content profile for the No. 308 pegmatite. The blue line is the diffusion profile after 20 years with $\beta = 0.08$. **d** Modeled and measured $\delta^7$Li profile for the No. 308 pegmatite. The blue line is the diffusion profile after 20 years with $\beta = 0.08$.

emplacement. Because pegmatites do not contain inherited phenocrysts[8], the initial pegmatite melts should approach the liquidus temperature of high-silica melts. The Macusani obsidian was considered to be a composition close to lithium pegmatite[38,39]. Its liquidus temperature is close to ~750 °C[39] at 200 MPa (the average pressure of pegmatite emplacement[8]) with 6 wt% H$_2$O content (the saturated H$_2$O content of rhyolitic melts at 200 MPa[40]). Therefore, we set the initial temperature of the pegmatite melt to 750 °C. Parameters for the pegmatite-forming melt are bulk density of 2300 kg/m$^3$, heat capacity of 1100 J/ kg*K, and thermal conductivity of 1.5 W/m*K[8,41].

The country rocks in the Jiajika deposit are schists and metamorphic sandstones, which have a bulk density of 2790 kg/m$^3$, specific heat of 1100 J/kg K[42]. Thermal conductivity of schists and metamorphic sandstones was obtained by λ (200 °C) = 0.75 + 705/(350 + 200)[42]. Pegmatite dikes were emplaced within the thermal aureole heated by their parent pluton[43]. Therefore, the country rock near the parent pluton has a high temperature, and the country rock far away from the pluton is close to the normal geothermal gradient. The crystallization temperature of many minerals in pegmatite is as low as ~ 375–475 °C[44], so 400 °C was chosen as the upper limit of the temperature of country rocks. Under normal geothermal gradient (i.e., ~20 °C/km), the rock temperature corresponding to 200 MPa is ~130 °C, so 100 °C was set as the lower limit of the temperature of country rocks. Most pegmatite dikes in nature are small, with widths of a few meters[45]. Very wide ones are the Tanco pegmatites that have an average thickness of ~40m[8]. In the thermal modeling, we therefore set the minimum and maximum values of the pegmatite width to be 2 m and 40 m (Fig. 5), respectively.

A representative thermal profile is shown in Fig. 5a. We simulated the intrusion of pegmatites with different widths into country rocks with different temperatures. Figure 5b–h correspond to different

country rocks with variable temperatures, respectively. It shows that when the country rock temperature is the same, thin pegmatite dikes have a shorter cooling time than wide dikes (Fig. 5i). When the dike width is the same, the higher the temperature of the country rock, the longer the cooling time of the pegmatite dikes. These observations are generally consistent with previous results[34]. Two factors influence the cooling time of pegmatite dikes: the temperature of country rocks at the time of pegmatite emplacement and the width of pegmatite dikes (Fig. 5i). Wider pegmatite dikes with higher country-rock temperature have longer cooling times compared with thinner pegmatite dikes with lower country-rock temperature. In addition, the initial temperature of pegmatite melt can also affect the cooling time of pegmatite dikes, with higher-temperature melts requiring longer cooling times. However, this factor is more complicated. Because the initial temperature of pegmatite melt is affected by melt composition, volatile content, and the content of fluxing components (such as Li and B)[46], and it needs to be considered in conjunction with phase equilibrium. This study set the same temperature of the initial pegmatite melt, but this does not affect our discussion of the influence of dike width and country-rock temperature because the effects of these factors on pegmatite cooling time are independent of each other.

## Country-rock temperature: a key control on Li-pegmatite genesis

Previous studies of the genesis of Li pegmatites have focused mainly on the origin of Li-rich melts, particularly the role of the crystallization process or melt–fluid immiscibility on Li enrichment[8,47,48]. This study, however, explores the transfer of Li after the emplacement of pegmatite dikes and the effect of this transfer on their ore-forming potential. There is clear evidence supporting the migration of Li from

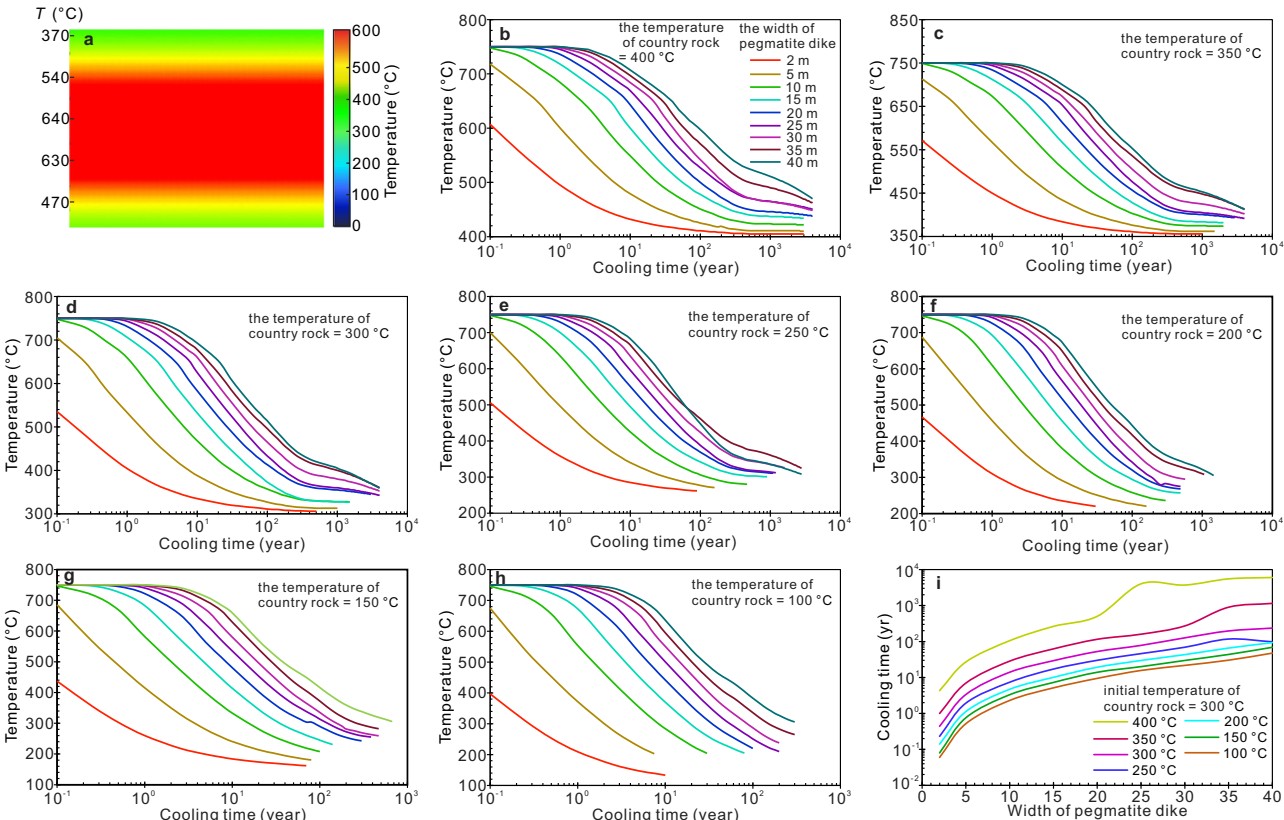

**Fig. 5 | Thermal modeling of the influence of the temperature of country rocks and the width of pegmatite dikes on the cooling time of dikes. a** A representative thermal profile of a pegmatite after emplacement. **b–h** Cooling processes of pegmatites with different widths and country rock temperatures. **i** Influence of pegmatite width on the cooling time of pegmatites within country rocks with different temperatures.

pegmatites into country rocks, as observed in the Jiajika pegmatite deposit (Fig. 3) and in previous studies[30,31,49]. This raises an important question as to whether this Li migration process affects the ore-forming grade of pegmatites. For this question, we carried out comprehensive thermal and diffusion modeling (Figs. 6 and 7).

Although this study analyzed the Li content of different pegmatites in Jiajika, the sample data are limited, and their Li contents are extremely heterogeneous. It is uncertain that the average value of these samples represents the true Li contents of the pegmatite dikes. To make our approach of broader significance, we use the statistical results provided by a previous study[50], i.e., Li-rich pegmatites, such as the No. 134 pegmatite in Jiajika, generally contain ~7000 ppm Li; zoned pegmatites, such as the No. 308 pegmatite in Jiajika, generally contain ~3500 ppm Li. First, we calculated the loss of Li in pegmatites only from the perspective of Li diffusion (Fig. 6). According to the diffusion equation, two key factors are affecting the diffusion loss of Li in pegmatite dikes that are the diffusion timescale and the dike width. Lithium diffusion from pegmatites into country rocks occurs only during the cooling of pegmatites and ceases when the pegmatites are completely crystallized. Therefore, the timescale of diffusion is also the cooling time of pegmatite dikes. Diffusion modeling of a Li-rich pegmatite (initial lithium content = 7000 ppm) show that its Li content in the pegmatite dike will decrease with increasing cooling time. We find that wider pegmatite dikes have a slower rate of Li loss (Fig. 6). For zoned pegmatites, such as the No. 308 pegmatite in Jiajika, we find that if it has an initial Li content of 3500 ppm, its Li content decreases to less than 2000 ppm after several decades of diffusion (Fig. 4a). In such a case, the pegmatite dike loses its ore-forming potential, as supported by the low-grade and tonnage of the No. 308 pegmatite.

Then, we use the cooling time as an input parameter (i.e., the results of the cooling time of pegmatite dikes with different widths and

country-rock temperatures), obtained from the thermal modeling, for the Li diffusion equation, resulting in the output values of the final Li contents and $\delta^7Li$ values of pegmatite dikes after diffusion (Fig. 7). We found a deviation from the expected findings: the final Li content of pegmatite dikes depends mainly on the temperature of country rocks, whereas the effect of the width of pegmatite dikes is negligible (Fig. 7a, c). Although the loss of Li by diffusion from a wider dike is slower (Fig. 6), its heat loss is also slower (Fig. 5), resulting in a longer cooling time (Fig. 5i). As a consequence, the ratio of Li loss to initial Li content for wide dikes is not much different from that of thin dikes (Fig. 7a).

In summary, a crucial finding of this study is that the Li content of a pegmatite dike (and whether the dike is economically viable) is controlled not only by the initial Li content of the pegmatite melt (which is influenced strongly by the degree of crystal fractionation[51], fluid exsolution[52,53]) but also by the temperature of host country rocks at the time of emplacement (Fig. 8). The lower the temperature of the country rocks, the more favorable the conditions for Li mineralization (Fig. 7c). This finding is consistent with the classic regional zoning pattern in a pegmatite field, whereby the pegmatites distal from the parent pluton are rich in Li, whereas the proximal pegmatites are generally barren in Li. All pegmatite dikes were emplaced within the thermal aureole heated by their parent pluton[8,43,54]. Country rocks near the plutons have high temperatures whereas distal rocks were at lower temperatures, as evidenced by the Buchan metamorphic series around the parent pluton in Jiajika[28] as well as thermal metamorphism in other pegmatite districts[55]. The farther away from the parent pluton and the lower the temperature of the country rocks, the lesser the loss of Li by diffusion from the pegmatite dike. For proximal pegmatites, the temperature of country rocks is high and the cooling time of the pegmatite dikes is long, meaning that most of the Li in the pegmatite bodies is lost

through diffusion (Fig. 8e, f). Another piece of evidence supporting our conclusion is the contrasting Li isotopic compositions of Li-poor and Li-rich pegmatites. In the Jiajika deposit, Li-rich pegmatites have lower $\delta^7$Li values compared with Li-poor pegmatites, and Li-poor pegmatites show wide variations in Li isotopic composition and many samples have high $\delta^7$Li values (up to +7.6‰; Fig. 2). This pattern is consistent with results from thermal and diffusion modeling, which show that when pegmatite dikes intrude high-temperature country rocks, they lose most of their Li and have elevated $\delta^7$Li values (Fig. 7c, d). A similar pattern (i.e., pegmatite samples with elevated $\delta^7$Li are derived mainly from Li-poor dikes) can also be seen in other pegmatite districts[15,56–58]. An exception is the Black Hills pegmatite deposit[59], possibly because of its large size (>150 km²). In this large area, different pegmatite swarms may have originated from different magmatic systems[60], and their initial pegmatite melts may have had different Li contents and isotopic compositions.

## Methods

### Whole-rock major- and trace-element analyses

Major element analyses of whole-rock were conducted at the Wuhan SampleSolution Analytical Technology Co. Ltd., Wuhan, China, using a Zsx Primus II wavelength dispersive X-ray fluorescence spectrometer (XRF) produced by RIGAKU, Japan. Sample powder (200 mesh) was placed in an oven at 105 °C for drying for 12 hours. 1.0 g sample was weighed and placed in the ceramic crucible and then heated in a muffle furnace at 1000 °C for 2 hours. After cooling to 400 °C, this sample was placed in the drying vessel and weighted again in order to calculate the loss on ignition (LOI). 0.6 g sample powder was mixed with 6.0 g cosolvent ($Li_2B_4O_7$:$LiBO_2$:LiF = 9:2:1) and 0.3 g oxidant ($NH_4NO_3$) in a Pt crucible, which was placed in the furnace at 1050 °C for 15 min. Then, this melting sample was quenched with air for 1 min to produce flat discs for the XRF analyses. The X-ray tube is a 4.0Kw end window Rh target, with a voltage of 50 kV, a current of 60 mA. All major element analysis lines are kα. The standard curve was established suing the national standard material (China), including rock standard sample GBW07101-14, soil standard sample GBW07401-08, stream sediment standard sample GBW07302-12. The data were corrected by

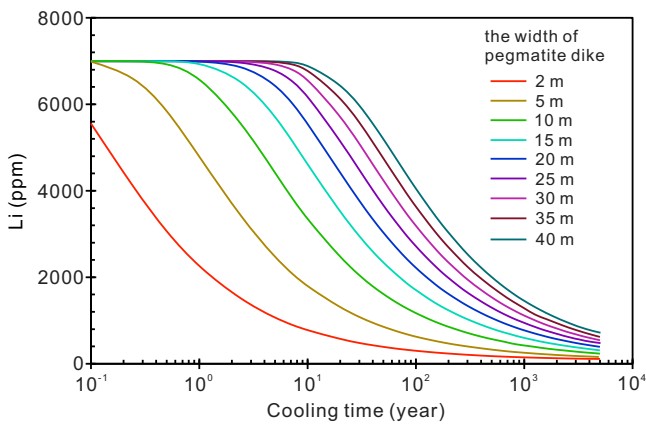

**Fig. 6 | Influence of cooling time on the final Li content of pegmatites with different widths.** The calculation of Li loss from pegmatites was from the perspective of Li diffusion. Obviously, the loss of Li by diffusion from wider dikes is slower than thinner dikes.

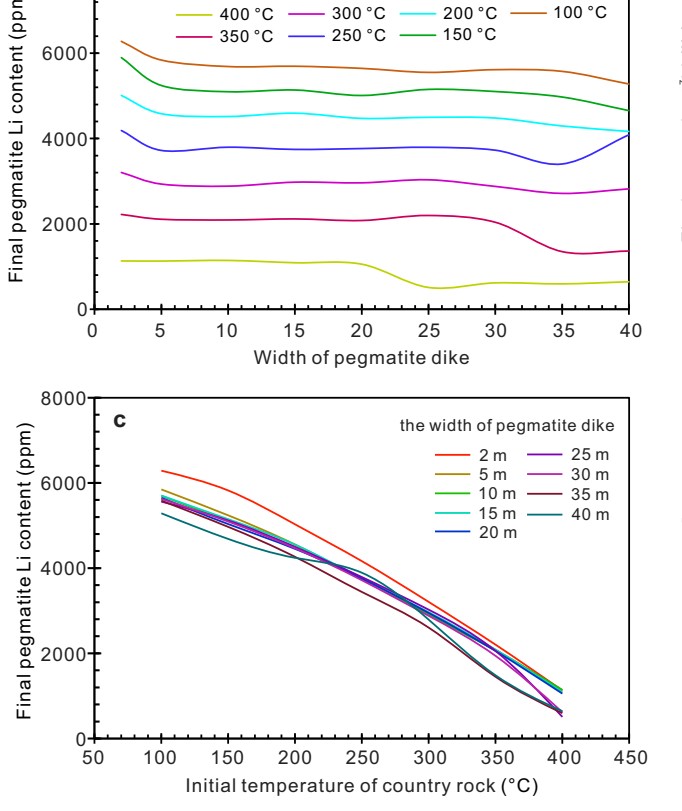
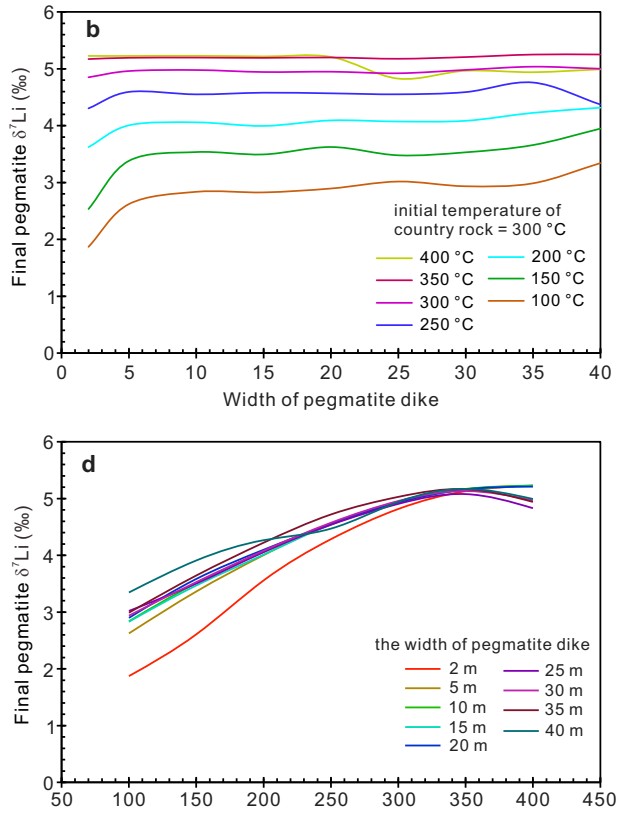

**Fig. 7 | The effect of dike width and country rock temperature on the final Li content and $\delta^7$Li value of pegmatite dikes, using a combination of thermal and diffusion modeling. a** Relationship between pegmatite dike width and final Li content. **b** Relationship between pegmatite dike width and final $\delta^7$Li values. **c** Relationship between the initial temperature of country rocks and final Li content of the pegmatites. **d** Relationship between the initial temperature of country rocks and final $\delta^7$Li values of the pegmatites.

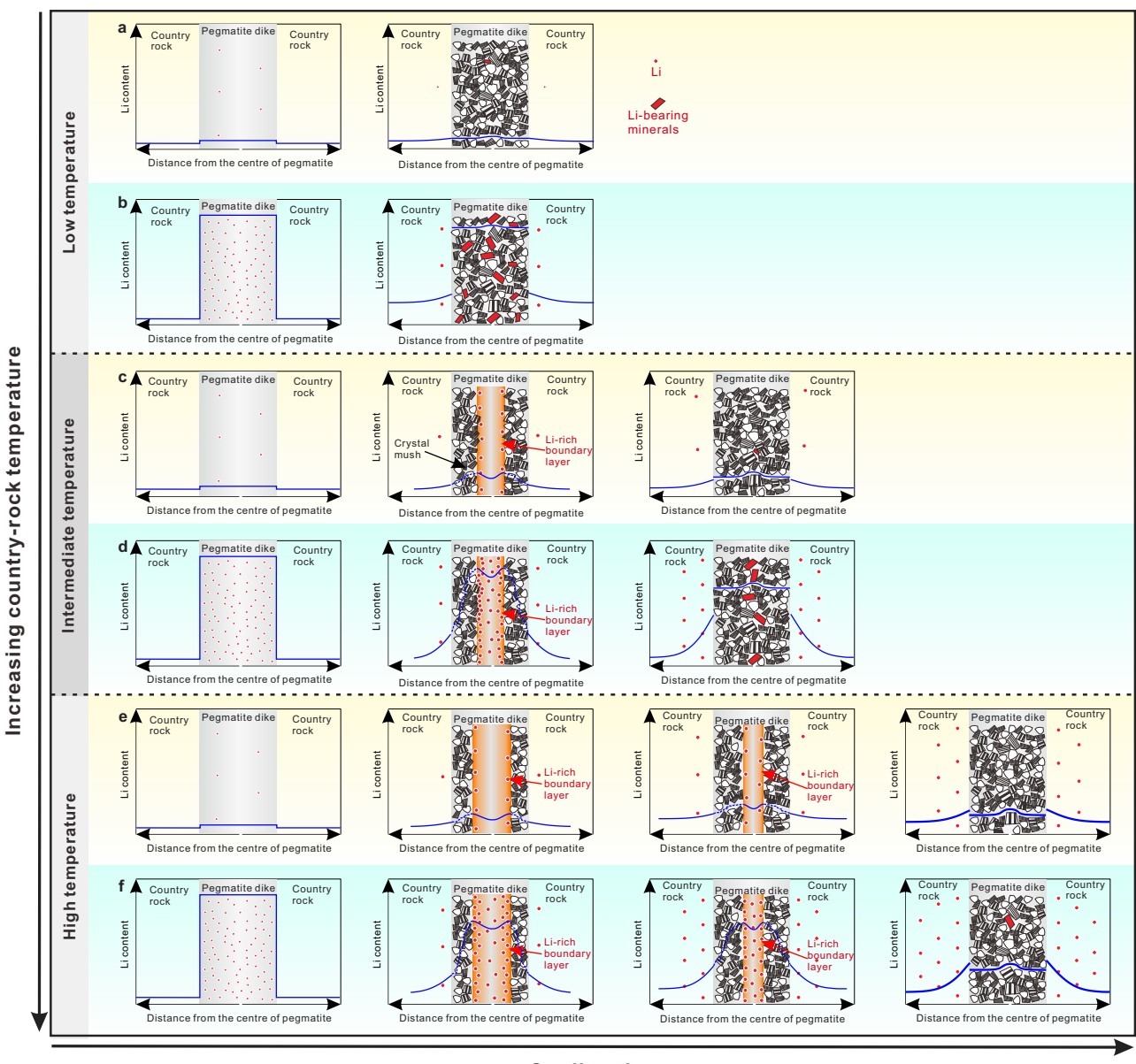

**Fig. 8 | The diffusion loss of Li when pegmatites intrude into country rocks that have different temperatures.** We considered two situations where the initial pegmatite melt was Li-poor (**a**, **c**, and **e**) and Li-rich (**b**, **d**, and **f**). In low-temperature country rocks (**a**, **b**), pegmatites cool very quickly, and the timescale for effective diffusion of Li is very short. Whether initial pegmatite melts are Li-rich or Li-poor, the amount of Li loss is very limited. In high-temperature country rocks (**e**, **f**), pegmatites cool very slowly, and it takes a very long timescale for Li diffusion. Even if the initial pegmatite melt is Li-rich, most of Li will migrate into the country rocks through diffusion. In the Jiajika deposit, the No. 134 pegmatite corresponds to the scenario of **b**; the No. 308 pegmatite corresponds to the process of **d**; the No. 433 pegmatite may correspond to the scenario of **e** or **f**. The blue lines are not a quantitative result of diffusion modeling, but an approximate illustration of the effect of diffusion on Li content changes.

theoretical α coefficient method and the relative standard deviation (RSD) is less than 2%.

Trace element analysis was also carried out at the Wuhan SampleSolution Analytical Technology Co. Ltd, using Agilent 7700e inductively coupled plasma mass spectrometry (ICP-MS). Sample powder (200 mesh) was placed in an oven (105 °C) for drying more than 12 hours. Then 50 mg sample powder was placed in a Teflon bomb and 1 ml $HNO_3$ and 1 ml HF were added into the Teflon bomb carefully. The bomb was installed in a stainless-steel pressure jacket, and the whole jackets were heated to 190 °C in an oven >24 hours. After they cooled, we opened the Teflon bombs and set them on a hotplate at 140 °C, evaporated to incipient dryness. Then 1 ml $HNO_3$ was injected and evaporated till dry. After that, 1 ml $HNO_3$, 1 ml Milli-Q water and

1 ml internal standard solution (1 ppm In) were injected, and the bombs were resealed and installed in an oven at 190 °C more than 12 hours. The solution was put into a polyethylene bottle and added 2% $HNO_3$ to 100 g for ICP-MS analysis. The RSD is <5%.

## Li isotope analyses

Lithium isotope analyses were conducted at the State Key Laboratory of Isotope Geochemistry (SKLaBIG), Guangzhou Institute of Geochemistry, Chinese Academy of Sciences (GIGCAS), Guangzhou, China, using the procedures described by Zhu et al.[61]. The weighed samples (2–60 mg, according to the Li contents) were added into cleaned 7 mL PFA Savillex® beakers and then digested with a 2:1 mixture of concentrated HF and 8 M $HNO_3$ on a hotplate for 7 days. After evaporating

till dry, aqua regia, concentrated $HNO_3$ and 6 M HCl were injected and evaporated to remove residual fluorides and complete digestion before residues were redissolved in 1 mL of 2.5 M HCl for ion-exchange chromatography. Lithium was purified using a single column loaded with AGMP-50 resin (BioRad™; 200–400 mesh), using a mixture of acid of 0.2 M HCl and 0.3 M HF, and 0.73 M HCl as eluents. The total Li procedural blank was lower than 0.2 ng. It is negligible compared to the 110–270 ng of Li which was loaded on the column. Isotope analyses were performed on a Thermo-Fisher Scientific Neptune plus MC–ICP–MS and instrumental mass fractionation was corrected using the standard-sample bracketing procedure. Lithium isotope data are reported as $\delta^7Li$ values relative to L-SVEC[62], expressed as $\delta^7Li$ (‰) = [$(^7Li/^6Li)_{sample}/(^7Li/^6Li)_{L\text{-}SVEC}-1$] × 1000. The standard deviation based the long-term external repeatability of $\delta^7Li$ for the standard solution is approximately ±0.52‰ (2 SD). Two rock standards (JG-2 and BHVO-2) and four replicate samples were analyzed to monitor data quality. The $\delta^7Li$ values for JG-2 (Granite, GSJ) and BHVO-2 (Basalt, GSJ) were −0.2 ± 0.1 ‰ and +4.8 ± 0.1‰ (2SE), respectively, consistent with the published values within 2 SD uncertainties (see summarized by Zhou et al.[15]). The variations of the measured $\delta^7Li$ values in duplicate samples were <0.7‰, indicating that the Li isotope compositions of the samples investigated in this study are homogenous and reproducible.

## Diffusion modeling

Previous models for Li diffusion from pegmatite into country rocks were based on the assumption that the pegmatite is an infinite lithium reservoir relative to the country rocks[12,13]. Here, we explored the influence of diffusion on pegmatite lithium contents and Li isotopic compositions, consistent with the approach in Zhou et al.[15]. An extended source model[63] was adopted to account for Li diffusion after the emplacement of pegmatite dikes. Assuming that when a pegmatite dike of $\phi$ meters in width intrudes country rocks, it can be regarded as a summation of point plane sources. The mass density of a single plane $\xi \in (-\phi/2, \phi/2)$ can be expressed as $C_0 \, d\xi$. The concentration contribution of this single plane to position $x$ is

$$\frac{C_0 d\xi}{(4\pi Dt)^{1/2}} e^{-(x-\xi)^2/4Dt} \qquad (1)$$

Total contribution of all the plane sources to position $x$ is given by

$$C(x,t) = \int_{-\phi/2}^{\phi/2} \frac{C_0}{(4\pi Dt)^{1/2}} e^{-(x-\xi)^2/4Dt} \, d\xi \qquad (2)$$

and the final solution is

$$C(x,t) = \frac{C_0 - C_1}{2} \left[ erf \frac{x+\phi/2}{2\sqrt{Dt}} - erf \frac{x-\phi/2}{2\sqrt{Dt}} \right] + C_1 \qquad (3)$$

where $x$ = distance from the center of the pegmatite dike, $t$ = duration of the diffusion process, $C(x,t)$ = element abundance at position $x$ after diffusion duration of $t$, $C_0$ = initial element abundance of the pegmatite dike, $C_1$ = element abundance in original country rocks, $D$ = diffusion coefficient, and erf = error function. The $^6Li$ and $^7Li$ are considered as two different elements, and their effective diffusion coefficients ($D$) conform to the relationship: $D_6/D_7 = (m_7/m_6)^{\beta}$ [64], where $\beta$ is an empirical parameter. Experimental studies reveal that $\beta$ value for water is <0.071[11] and 0.215 for silicate melts[10]. The $\beta$ value yielding the best fit to the measured profiles in Jiajika is 0.08. For the Jiajika country rocks we assumed $D = 10^{-7}$ $m^2 s^{-1}$, consistent with other fluid-assisted grain-boundary diffusion models[12,13]. The fluid assisting diffusion here may be pegmatite-derived fluids (in areas close to the pegmatite), or it may be free water carried in the grain boundaries or cracks of low-grade metamorphic rocks (in areas farther away from the pegmatite). It is likely that Li contents and isotopic compositions of country rocks

adjacent to the pegmatites were modified during the emplacement of the pegmatites[15]. We assumed that the far-field country rocks represent the background values of country rocks (average Li = 78.2 ppm; average $\delta^7Li$ = 0.23). The No. 308 pegmatite in Jiajika is a zoned pegmatite body, with a width on the surface is ~30m[65]. This zoned pegmatite generally contains ~3500 ppm Li[50]. Lithium-rich pegmatites, such as the No. 134 pegmatite in Jiajika, generally contains ~7000 ppm Li[50]. We assumed an initial $\delta^7Li$ value of −0.5‰ for the pegmatite melts that is the average value of the No. 134 pegmatite, because it experienced minimal kinetic fractionation and its average value should be closest to the initial value of pegmatite melts.

## Data availability

All data reported in this manuscript are included in the Table 1 and Supplementary Data 1, and they have been deposited in the Figshare repository (https://doi.org/10.6084/m9.figshare.27824010).

## Code availability

The HEAT3D program is available at https://www.lanl.gov/orgs/ees/geodynamics/Wohletz/KWare/Index.htm.

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

## Acknowledgements

Q. W. was supported by the National Natural Science Foundation of China (42021002). J.S. Z was supported by Strategic Priority Research Program (A) of the Chinese Academy of Sciences (XDA0430102), the National Natural Science Foundation of China (42372067), and Tuguangchi Award for Excellent Young Scholar GIGCAS (TGC202202). This is contribution No. IS-3600 from GIGCAS. We thank Bao-Di Wang, Yun-Feng Xu and Jin-Heng Liu for field help.

## Author contributions

Q. W. initiated this study. Q. W. and J.S. Z. conceived and designed the project. Q. W., J.S. Z., and H. W. performed field work. J.S. Z. performed chemical analyses, thermal and diffusion modeling. J.L. M., G.H. Z. and L. Z. contributed to Li isotope analyses. J.S. Z. and Q. W. interpreted the data and wrote the original draft. All authors contributed to improving the manuscript.

## Competing interests

The authors declare no competing interests.
