## [Peer Review File · Nature Communications]

Reviewers' comments:

Reviewer #1 (Remarks to the Author):

Please see attached PDF file.

Reviewer #2 (Remarks to the Author):

We have reviewed this manuscript entitled: "Pegmatite lithium deposits formed within low-temperature 1 country rocks" written by Zhou et al. The supply of Li for electric vehicles has a great risk in the next years, and thus it is critical to decode the nature of pegmatite and their implications for exploration. In this study, the authors present a detailed Li isotopes from three distinct pegmatites at Jiajika to provide the new constraints on hard-rock type Li deposits. They suggest a novel interpretation that the low temperature country rock is the crucial parameter for rapid ore formation. This manuscript is well organized. However, some parts need further modification before publications. I recommend a major revision on it.

General concerns:

The authors suggest that the country-rock profile line of the No. 308 pegmatite can be well explained by the diffusion model of Li via the variations of Rb-Li contents and Li isotopes. However, the trends of the 134 pegmatite is inconsistent with model. The authors stress that the gradually increasing Li content on the right should be affected by another pegmatite dike, but it is not convinced for me. From the geologic map, the size of the 134 pegmatite is clearly bigger than the nearby dikes, and thus, it is hard to believe that the unknown little sized dike shows stronger effects of diffusion.

This paper emphasizes that temperature-controlled lithium diffusion in surrounding rocks affects the lithium mineralization of pegmatites, so what are the conditions for lithium diffusion? and why is there almost no diffusion effect in 134 veins in this study.

The diffusion for the Li-poor 433 pegmatite vein is unreliable. Firstly, following the fractionation model, the Li-poor pegmatites formed in the early stage, and they are generally featured by low concentrations of Li. It does not need the diffusion model to explain the low Li in this pegmatite. Moreover, the high concentrations of wall rocks have been reported, and the diffusion model is unnecessary for this phenomenon. Thirdly, if the diffusion occurred in 433 pegmatite vein, but why it did not occur in Li-rich 134 and 308 veins. Lastly, the data of NO.433 vein should also be given in the paper.

In this contribution, the authors emphasized the critical role of the temperatures of wall rocks in the formation of pegmatite Li deposit. However, the authors do not give any data of the temperature of wall rocks at the Jiajika deposit when the pegmatite emplaced. The depth of these wall rocks is related to their temperatures considering regions geothermal gradient. So what is the emplacement depth may provide the clues for temperature of wall rocks.

The authors considered the temperature of wall rocks as main contributor to the cooling time of pegmatite dikes. But to the best of our knowledge, pegmatites were formed under supercooling conditions, which means that the composition of the pegmatitic melts may be influenced by the cooling time. Please explain the validity of your theoretical model in the geologic conditions.

The mechanism of lithium isotope fractionation is not clear. Are there any other mechanisms (e.g. source heterogeneity, supercritical fluid) besides fluid metasomatism and diffusion to make Li isotope variation?

The natural case of the Jiajika deposit is good. Can you give more specific natural cases to support your idea that cooling time can severely affect the formation of the Li deposit? If it is a common phenomenon for rare metal pegmatite, then the new model would be important.

Specific concerns:

The granite name is "Majingzi" not "Majinzi". Please check the text, tables, and figures.

Line 119, please clarify that lithium isotopic compositions of the No. 134 pegmatite are homogeneous.

Review: Manuscript Number: NCOMMS-23-55961

Title: Pegmatite lithium deposits formed within low-temperature country rocks

The study by Zhou et al. explores the transfer of Li after the emplacement of pegmatite dikes and the effect of this Li transfer on their ore-forming potential. The authors try to answer the question if Li migration processes affect the ore-forming grade of pegmatites. To try answer this question they present thermal and diffusion modelling. The authors conclude that Li-rich pegmatites form when the melt intrudes country rock that has a low temperature. The abstract highlights very well the importance of Li for our society and why it is necessary to understand Li behaviour to guide Li exploration, however, this takes up most of the abstract and could be significantly shortened to give more room for the actual study where the authors could provide data that is linked to their major findings. Similarly, in the introduction the authors highlight the relevance of Li to achieve net-zero CO₂ emissions, however, the authors fail to introduce the findings of other relevant studies that have focused on Li diffusion between country rocks and pegmatites, e.g., the pioneering study by Teng et al. (2006), and Marks et al. (2007), Liu et al. (2010) or their own publication Zhou et al. (2021). The authors move on to the migration of Li from pegmatite into country rocks by diffusion. Here the authors show their findings which are very similar to the ones found by Teng et al. (2006) who investigated the Li diffusion from pegmatite into country rocks of the Tin Mountain pegmatite. In this part the manuscript could be improved by putting their findings in the context with the findings of previous studies and the authors could highlight why their findings are innovative compared to previous studies. Further, it is not clear which kind of Li diffusion the authors envision: fluid- assisted grain-boundary diffusion or solid-state diffusion. The data and interpretations throughout the text are not sufficiently put in context and previous work of other authors is not sufficiently considered and the applied diffusion and thermal models need to be explained in more detail. Figures 4 and 5 for example need more explanation to support the statement that Li-rich pegmatites only form when melt intrudes low-temperature country rock. The thermal modelling which is one of the key points of this manuscript is not explained in detail, it is not clearly explained how the authors picked the parameters of their models, e.g., boundary conditions, temperature. Errors on their trace elements and lithium isotopic measurements are not provided in table 1 and errors on their modelling are also not provided or explained. Further, I recommend to restructure the paper and have a clear results part and a discussion part, at the moment results and discussion are mixed which in parts makes it difficult to follow. Below I outline my minor and major comments that I hope will help to improve the manuscript. I believe this paper could be a great contribution to our knowledge, however, I believe that major revisions are necessary.

Minor comments:

Line 73: but **also** on

Lines 83-84: please add values for the Li abundance – what do you consider here as high abundance compared to the Li-poor dikes?

Line 106: please provide the Li value for dike 433

Line 119: please clearly state here that you are talking about dike 134 and the country rock

Line 150: compared **to** the

Line 233: evaporated **till dry**

Line 244: After **evaporating till dry**

Line 126-129: There are no references provided to support the statement that the Rb profile can be used to identify if fluids were involved. Additionally, the partitioning behaviour of Rb may vary compared to Li therefore the Rb profile might be shorter compared to the Li profile. Therefore, without any reference that support this statement this is not a valid argument to exclude the influence of fluids over longer distances (20 m). Please provide references and/or explain the involvement of fluids in more detail.

Line 143: here it is stated that crystallisation in pegmatites is rapid – please add references, and further explain what you mean with rapid in this context. Do the authors mean with rapid a few days or decades. Please clarify.

Line 146: “although the Li content of the entire pegmatite dike is low at this time (Fig. 3).” I am not sure what the authors mean by that, is the Li concentration of the pegmatite low to start with? Please clarify.

Line 298: why is -0.5 ‰ the average for the granites (average would be -1.25 ‰ when using the authors data), please clarify.

No analytical errors are provided for either the trace element measurements or the $\delta^7\text{Li}$ values. They are not presented in table 1, in the figures or the supplement file. Please provide the analytical errors on your measurements as these might affect the interpretation and/or your diffusion models.

Major comments:

The introduction does not provide any information on previous work on Li diffusion in pegmatites or how their work is different compared to previous studies. I recommend to add a short summary of the findings of previous studies, e.g., what did they find and how did they explain it.

I recommend to add information about the samples, this could be provided in the supplementary material. For example, which minerals are present and to which percentage are they present? The mineral abundance will have an effect on the isotopic composition of the bulk sample. Also mention in the text how the samples were collected – in which distance from the contact. They are presented in

table 1 but I would recommend to add an extra sentence. This could be added to the text around Line 108.

Throughout the main text it is not clearly stated what kind of diffusion the authors talk about. In line 132 to 133 they state that the fluid only worked over a short distance, it would help the reader and benefit the manuscript if the authors explained what kind of diffusion they are referring to either fluid- assisted grain-boundary diffusion or solid-state diffusion. It would help the manuscript if the authors provided arguments/explanations for why they prefer one over the other and how their data, e.g., the Rb profile support this statement.

In line 136 to 137 the authors explain that their data confirms that Li diffuses from the pegmatite into the country rock. The way this sentence is phrased it makes it sound like this is a new finding, however, the study by Teng et al. 2006 already showed this, please add a reference here. Further, it is not clear why only dike 308 was modelled or why the dike was chosen for the model. I assume it is based on the concentration and isotope profiles, please clarify this in the text.

The authors show that their concentration and isotope profile must have formed within 35 years and state that this confirms that diffusion must have driven this migration of Li. This does not confirm Li diffusion; the authors are using a circular argument here. The fact that it continuously decreases in concentration and the large fractionation of the isotopes confirm that it is a diffusion profile. Please rephrase this part of the text. State clearly that due to this profile a diffusion model is applied to the data, this diffusion model reveals that the Li concentration and $\delta^7\text{Li}$ profile must have formed within 35 years. If the authors mean that their result of 35 years confirms that it is diffusion this needs to be explained. Additionally, I think it would improve the paper if the authors could put their result in context with others papers that focus on cooling rates of pegmatites, e.g., Simmons et al. (2008), Webber et al., (1999), Phelps et al. (2020). Cooling rates of pegmatites are a strongly debated topic and the findings of the authors could add valuable information to the discussion.

Line 138 to 150 and Figure 3. Based on the data presented by the authors I do not understand how the authors arrive at the conclusion that the pegmatite lost all Li to the country rock resulting in an enrichment of the country rock and a full depletion of the pegmatite. Do the authors assume the melt was already low and all the Li is leached from the dike? Several measurements of the Schist that is related to dike 134 have similarly high concentrations in Li. I would recommend to find more supportive arguments for why Li should diffuse completely out of the dike into the country rock, what is the difference between the dikes or the country rock that this would happen? Are other minerals present in the country rock around this dike compared to dike 308 that Li preferentially partitions into the host rock? Are you assuming all Li is leached from the pegmatite into the country rock by fluids? This part of the text needs to be expanded and explained in more detail.

It is not clear if the blue line in figure 3 is supposed to be the diffusion profile for the $\delta^7\text{Li}$ values, please add this information to the figure caption. If this is the diffusion profile for the $\delta^7\text{Li}$ values, how did the authors model this profile? Is it based on figure 2h? If so why would this be valid, since both dikes 433 and 308 behave differently. Throughout the text it is also not explained why these dikes are so different and how one dike lost all its Li whereas the other did not, it would greatly improve the manuscript if the authors further expanded on this.

Figure 2 – I believe this figure could be improved by 1) giving a, b and c the same scale so they are comparable and give d, e and f the same scale; 2) I would combine b and c with each other and e and f with each other and instead show the data for the third dike in c and f; 3) I believe 2g and 2h should be a separate figure. In the figure caption I would also mention that for the Li concentrations of the dikes the average values are used or the measured data could be plotted in that figure as well. It is not clear and also not explained why the authors model for 10, 50 and 100 years and why in 2g and 2h no model for 35 years is shown. Please give further information in the text. It is also not clear why the authors use 3500 ppm in 2g) as the initial concentration of pegmatite 308 since the concentration of this dike was measured, why is literature data used here? Similarly, in 2c) the boundary condition that are applied by the authors are not clearly explained and it is not explained why these boundary conditions are applied. Same is true for the isotope models. Please clarify in the text. How would different diffusion coefficients influence the results and why was this specific diffusion coefficient applied? The partitioning of Li depends on the minerals that are present in the host rock and in the pegmatite however this issue is not addressed at all in the text or the methods, please explain further in the text.

In the section “Country-rock temperature: a key control on Li pegmatite genesis” the authors refer to their figure 4 and 5, however the authors do not explain their figures. It is not explained why these temperatures for the country rock were chosen, why the different widths for the dikes were chosen. It would improve the manuscript if the authors explained their figures in more detail and how they chose the values for the parameters. How does the model change from 4b to 4h? What can we learn from that? How does it affect the cooling time? Why was 750 °C used as the starting temperature, how would other starting temperatures affect this model? In the text it says 3500 ppm was chosen as the boundary condition for this model, why and also in Figure 5a the boundary condition is more around 7000 ppm? Are the authors modelling different dikes? This is not clear from the figure caption or the main text, please clarify in the text. The authors state that the width of the dikes places a role for the cooling time, this has been addressed by others studies before, please reference them and also how do the findings in this study compare to other studies? Why are the concentrations in Figure 5 different to Figure 2? Figure 5 is not sufficiently explained in the text, please give more information. In the thermal modelling section, the authors could give more explanation for the values they use for their models.

Response to Referees Letter

Response to Reviewer #1

Reviewer #1 (Remarks to the Author):

Review: Manuscript Number: NCOMMS-23-55961

Title: Pegmatite lithium deposits formed within low-temperature country rocks

The study by Zhou et al. explores the transfer of Li after the emplacement of pegmatite dikes and the effect of this Li transfer on their ore-forming potential. The authors try to answer the question if Li migration processes affect the ore-forming grade of pegmatites. To try answer this question they present thermal and diffusion modelling. The authors conclude that Li-rich pegmatites form when the melt intrudes country rock that has a low temperature. The abstract highlights very well the importance of Li for our society and why it is necessary to understand Li behaviour to guide Li exploration, however, this takes up most of the abstract and could be significantly shortened to give more room for the actual study where the authors could provide data that is linked to their major findings. Similarly, in the introduction the authors highlight the relevance of Li to achieve net-zero CO₂ emissions, however, the authors fail to introduce the findings of other relevant studies that have focused on Li diffusion between country rocks and pegmatites, e.g., the pioneering study by Teng et al. (2006), and Marks et al. (2007), Liu et al. (2010) or their own publication Zhou et al. (2021). The authors move on to the migration of Li from pegmatite into country rocks by diffusion. Here the authors show their findings which are very similar to the ones found by Teng et al. (2006) who investigated the Li diffusion from pegmatite into country rocks of the Tin Mountain pegmatite. In this part the manuscript could be improved by putting their findings in the context with the findings of previous studies and the authors could highlight why their findings are innovative compared to previous studies. Further, it is not clear which kind of Li diffusion the authors envision: fluid- assisted grain-boundary diffusion or solid-state diffusion. The data and interpretations throughout the text are not sufficiently put in context and previous work of other authors is not sufficiently considered and the applied diffusion and thermal models need to be explained in more detail. Figures 4 and 5 for example need more explanation to support the statement that Li-rich pegmatites only form when melt intrudes low-temperature country rock. The thermal modelling which is one of the key points of this manuscript is not explained in detail, it is not clearly explained how the authors picked the parameters of their models, e.g., boundary conditions, temperature. Errors on their trace elements and lithium isotopic measurements are not provided in table 1 and errors on their modelling are also not provided or explained. Further, I recommend to restructure the paper and have a clear results part and a discussion part, at the moment results and discussion are mixed which in parts makes it difficult to follow. Below I outline my minor and major comments that I hope will help to improve the manuscript. I believe this paper could be a great contribution to our knowledge, however, I believe that major revisions are necessary.

We are very grateful to Reviewer #1 for providing constructive, insightful and detailed reviews of this manuscript! The comments led to improvements of the manuscript significantly. Thanks for your time and patience!

In the revised manuscript, we have made the following changes:

- (1) The abstract part has been shortened, as: “The global climate crisis is likely to lead a potential

supply risk of lithium (Li) over the coming decades. More than half of the world's production of Li is derived from Li-bearing pegmatites. Although pegmatites are widespread, only a small fraction host Li mineralization. Revealing which factors cause some pegmatites to be enriched in Li and others to be barren is critical for understanding Li pegmatite formation and for guiding exploration for new Li resources. In this study, we used an innovative approach involving the analysis of natural samples from the world's largest pegmatite Li deposit (Jiajika, China). Here we show a surprising result that Li contents in pegmatites are controlled not only by the initial Li contents in pegmatite melts but also by the temperature of country rocks at the time of pegmatite emplacement. Lithium-rich pegmatites form preferentially when Li-rich pegmatite melts intrude low-temperature country rocks." Since Nature Communications requires that the abstract cannot exceed 150 words, the shortened abstract has 147 words, so it is difficult for us to add more result descriptions in the abstract part.

(2) In the introduction part, we have added a brief review of the relevant content of this study, especially the literatures you mentioned (in fact, most of these literatures have been cited in the first draft), and introduced the differences between this study and previous work, as "Although Li is a metallic element, its atomic mass is very small (6.941), only higher than H (1.008) and He (4.003). Together with its low ionic charge and size, Li has very high mobility. Therefore, Li can easily diffuse during high-temperature geological processes⁹. Diffusion process can be well recorded by Li isotopes. Lithium has two stable isotopes (i.e., ⁶Li and ⁷Li), and ⁶Li has a higher diffusion rate than ⁷Li (about 3% faster)^{10,11}. As a result, diffusion will form a unique Li isotope profile. Lithium isotope diffusion profile has been found around lithium pegmatites as well as granitic plutons^{12,13,14}, confirming that lithium diffusion occurs during the emplacement of pegmatites. However, the purpose of these studies was to reveal the kinetic fractionation behavior of lithium isotopes and did not pay attention to the effect of diffusion on Li content as well as Li mineralization of pegmatites^{12,13}. Moreover, pegmatites were assumed to be an infinite element Li reservoir in these studies^{12,13}, meaning that diffusion has no effect on Li content as well as mineralization ability of pegmatite itself. Based on an observation of substantially higher $\delta^7\text{Li}$ values of Li-poor pegmatites than Li-rich dikes, a recent study attributed the higher $\delta^7\text{Li}$ values of Li-poor pegmatites to diffusion-driven fractionation with a long duration, and speculated that diffusion may decrease the Li content of pegmatites¹⁵. Such a process of Li loss is critical for the formation of pegmatite Li deposit because it can determine the final grade and tonnage of pegmatite ore bodies. However, what factors account for the degree of Li loss by diffusion during pegmatite emplacement remains unclear."

In addition, we would like to explain a special point here that previous studies on lithium diffusion around pegmatites focused on the fractionation mechanism of lithium isotopes. These studies did not explore whether the diffusion process would affect the mineralization ability of pegmatites. Moreover, these studies all assumed that the pegmatite is an infinite lithium reservoir, which means that diffusion will not change the lithium content as well as lithium mineralization ability of pegmatites. As far as we know, our work is the first study to explore how diffusion affects the mineralization ability of pegmatites based on lithium element and isotope profiles around pegmatite dikes. More importantly, we attempt to present a new concept that pegmatite lithium deposits are preferentially formed within low-temperature country rocks, which has never been mentioned before (as far as we know).

(3) “fluid- assisted grain-boundary diffusion” has been added in the methods section, as “...consistent with other fluid-assisted grain-boundary diffusion models^{12,13}. The fluid assisting diffusion here may be pegmatite-derived fluids (in areas close to the pegmatite), or it may be free water carried in the grain boundaries or cracks of low-grade metamorphic rocks (in areas farther away from the pegmatite).”

(4) For thermal and diffusion modeling, we have added more detailed descriptions in the revised manuscript:

“Timescale of pegmatite cooling

Obviously, the amount of Li migration from the pegmatite dike into country rocks depends on the duration of the cooling of pegmatites. The cooling time of pegmatite dike can be constrained by thermal modeling^{8,35}. Here, a conductive cooling model (HEAT3D³⁸) was employed to track the thermal history of pegmatite dikes after their emplacement. Because pegmatites do not contain inherited phenocrysts⁸, the initial pegmatite melts should approach the liquidus temperature of high-silica melts. The Macusani obsidian was considered to be a composition close to lithium pegmatite^{36,39}. Its liquidus temperature is close to $\sim 750^{\circ}\text{C}$ ³⁹ at 200 MPa (the average pressure of pegmatite emplacement⁸) with 6 wt% H₂O content (the saturated H₂O content of rhyolitic melts at 200 MPa⁴⁰). Therefore, we set the initial temperature of the pegmatite melt to 750 °C. Parameters for the pegmatite-forming melt are bulk density of 2300 kg/m³, heat capacity of 1100 J/ kg*K, and thermal conductivity of 1.5 W/m*K^{8,41}.

The country rocks in the Jiajika deposit are schists and metamorphic sandstones, which has a bulk density of 2790 kg/m³, specific heat of 1100 J/kg K⁴². Thermal conductivity of schists and metamorphic sandstones was obtained by $\lambda (200^{\circ}\text{C}) = 0.75 + 705/(350 + 200)$ ⁴². Pegmatite dikes were emplaced within the thermal aureole heated by their parent pluton⁴³. Therefore, the country rock near the parent pluton has a high temperature, and the country rock far away from the pluton is close to the normal geothermal gradient. The crystallization temperature of many minerals in pegmatite is as low as $\sim 375\text{--}475^{\circ}\text{C}$ ⁴⁴, so 400 °C was chosen as the upper limit of the temperature of country rocks. Under normal geothermal gradient (i.e., $\sim 20^{\circ}\text{C}/\text{km}$), the rock temperature corresponding to 200 MPa is $\sim 130^{\circ}\text{C}$, so 100 °C was set as the lower limit of the temperature of country rocks. Most pegmatite dikes in nature are small, with widths of a few meters⁴⁵. A very wide one is the Tanco pegmatites that have an average thickness of $\sim 40\text{m}$ ⁸. In the thermal modeling, we therefore set the minimum and maximum values of the pegmatite width to be 2 m and 40 m (Fig. 5), respectively.

A representative thermal profile is shown in Figure 5a. We simulated the intrusion of pegmatites with different widths into country rocks with different temperatures. Figures 5b–5h correspond to different country rocks with variable temperatures, respectively. It shows that when the country rock temperature is the same, thin pegmatite dikes have a shorter cooling time than wide dikes (Fig. 5i). When the dike width is the same, the higher the temperature of the country rock, the longer the cooling time of the pegmatite dikes. These observations are generally consistent with previous results³³. Two factors influence the cooling time of pegmatite dikes: the temperature of country rocks at the time of pegmatite emplacement and the width of pegmatite dikes (Fig. 5i). Wider pegmatite dikes with higher country-rock temperature have longer cooling times compared with thinner pegmatite dikes with lower country-rock temperature. In addition, the initial temperature of pegmatite melt can also affect the cooling time of pegmatite dikes, with higher temperature melts

requiring longer cooling times. However, this factor is more complicated. Because the initial temperature of pegmatite melt is affected by melt composition, volatile content, and the content of fluxing components (such as Li and B)⁴⁶, and it needs to be considered in conjunction with phase equilibrium. This study set a same temperature of the initial pegmatite melt, but this does not affect our discussion of the influence of dike width and country-rock temperature, because the effects of these factors on pegmatite cooling time are independent of each other.”

“Although this study analyzed the Li content of different pegmatites in Jiajika, the sample data are limited and their Li contents are extremely heterogeneous. It is uncertain that the average value of these samples represents the true Li content of the pegmatite dikes. In order to make our approach of the broad significance, here we use the statistical results provided by the previous study⁵⁰, i.e., Li-rich pegmatites, such as the No. 134 pegmatite in Jiajika, generally contain ~7000 ppm Li; zoned pegmatites, such as the No. 308 pegmatite in Jiajika, generally contain ~3500 ppm Li. First, we calculated the loss of Li in pegmatites only from the perspective of Li diffusion (Fig. 6). According to the diffusion equation, two key factors affecting the diffusion loss of Li in pegmatite dikes are diffusion timescale and dike width. Lithium diffusion from pegmatites into country rocks occurs only during the cooling of pegmatites and ceases when the pegmatites are completely crystallized. Therefore, the timescale of diffusion is also the cooling time of pegmatite dikes. Diffusion modeling of a Li-rich pegmatite (initial lithium content = 7000 ppm) show that its Li content in the pegmatite dike will decrease with increasing cooling time. The wider the pegmatite dike, the rate of Li loss is slower (Figure 6). For zoned pegmatites, such as the No. 308 pegmatite in Jiajika, if it has an initial Li content of 3500 ppm, its Li content decreases to less than 2000 ppm after several decades of diffusion (Fig. 4a). In such a case, the pegmatite dike loses its ore-forming potential, as supported by the low grade and tonnage of the No. 308 pegmatite. Then, we input the cooling time (i.e., the results of the cooling time of pegmatite dikes with different widths and country-rock temperatures) obtained from the thermal modeling, into the Li diffusion equation, leading to the output of the final Li contents and $\delta^7\text{Li}$ values of pegmatite dikes after diffusion (Fig. 7). There is an unexpected result: the final Li content of pegmatite dikes depends mainly on the temperature of country rocks, whereas the effect of the width of pegmatite dikes is negligible (Fig. 7a and 7c). Although the loss of Li by diffusion from a wider dike is slower (Fig. 6), its heat loss is also slower (Fig. 5), resulting in a longer cooling time (Fig. 5i). As a consequence, the ratio of Li loss to initial Li content for wide dikes is not much different from that of thin dikes (Fig. 7a).”

(5) In the revised manuscript, analytical errors have been added in the methods part, as “The standard deviation based the long-term external repeatability of $\delta^7\text{Li}$ for the standard solution is approximately $\pm 0.52\%$ (2SD).” and “The relative standard deviation (RSD) is less than 5%.”. In addition, a sentence “The error bars are smaller than the symbol size because the standard solution of Li isotope analyses is approximately $\pm 0.52\%$ (2SD).” has been added in the captions of Fig. 2 and 3. For diffusion and thermal modeling, we did not know yet how to provide errors. In the relative publications, they also did not provide simulation errors (Webber et al., 1999; Teng et al., 2006).

(6) In the revised manuscript, the results and discussion parts are separated, and a description of the results has been added.

“Lithium isotopes and contents

Lithium elemental and isotopic compositions of the Li-rich pegmatites (i.e., the No. 134 pegmatite), Li-bearing pegmatites (i.e., the No. 308 pegmatite), Li-poor pegmatites (i.e., the No. 433 pegmatite), the Majingzi two-mica granites, the country rocks adjacent to the three pegmatites, and the far-field country rocks are listed in Table 1 and plotted in Figure 2. The Li content of the five pegmatite samples in the No. 134 pegmatite ranges from 6051 to 13720 ppm, but the changes in $\delta^7\text{Li}$ are small (-1.3‰ to $+0.1\text{‰}$). In the country-rock profile around the No. 134 pegmatite, the three adjacent samples have approximately identical Li contents and $\delta^7\text{Li}$ values (Fig. 3a and 3c). Starting from the ~ 3 -meters position, the Li content gradually increases outwards (Fig. 3a), while the $\delta^7\text{Li}$ generally remains unchanged ($+0.5\text{‰}$ to $+2.0\text{‰}$; Fig. 3c). The Li content of the No. 308 pegmatite fluctuates greatly (455 ppm to 19385 ppm), while the $\delta^7\text{Li}$ values are relatively uniform (-0.9‰ to $+1.7\text{‰}$). In the country-rock profile around the No. 308 pegmatite, the Li content gradually decreases (Fig. 3b), and the $\delta^7\text{Li}$ also gradually decreases simultaneously (Fig. 3d). An abnormal circumstance occurred in the No. 433 pegmatite: the pegmatite itself has very low Li contents (25 ppm to 92.3 ppm) but very high $\delta^7\text{Li}$ values ($+5.4\text{‰}$ to $+7.6\text{‰}$). In contrast, its adjacent country rocks have higher Li contents (894 to 1049 ppm). The Majingzi granites have a relatively identical Li contents (199 to 294 ppm) and $\delta^7\text{Li}$ values (-1.7‰ to -1.0‰ ; Fig. 2). The far-field country rocks (about several kilometers away from the Jiajika deposit) have lower Li contents (14.1 ppm to 128 ppm), and similar $\delta^7\text{Li}$ values (-1.9‰ to $+1.5\text{‰}$). We also present the literature data of the Jiajika deposit (Fig. 2), and the distribution patterns of Li content and Li isotopes are generally consistent with this study: (1) Li-rich pegmatites generally have lower $\delta^7\text{Li}$ values than Li-poor pegmatites; (2) Some Li-poor pegmatites have very high $\delta^7\text{Li}$ values; (3) The Li content and $\delta^7\text{Li}$ value of the country rocks in the deposit range greatly, and some samples have abnormally low $\delta^7\text{Li}$ values; (4) The Li content and $\delta^7\text{Li}$ of the Majingzi granites are relatively homogeneous.”

For other questions, please see point-by-point responses below.

Minor comments:

Line 73: but also on

Thanks. This problem has been modified.

Lines 83-84: please add values for the Li abundance – what do you consider here as high abundance compared to the Li-poor dikes?

Thanks. This sentence is quoted from the literature published by the exploration workers of the Jiajika deposit (Fu et al., 2023). It does not provide detailed information on the lithium content of these 30 pegmatites. The exploration workers should have conducted systematic sampling and lithium content analyses on most pegmatites. Compared with the industrial grades of Li deposits, they determined that these 30 pegmatites are Li pegmatites. So these pegmatites can be called lithium-rich pegmatites. However, the Li content data is an internal material of the exploration company and we cannot see it now.

Line 106: please provide the Li value for dike 433

Thanks. For the No. 433 pegmatite, there is no associated publication. In the revised manuscript, the lithium content of the No. 433 pegmatite is presented in the results section, as “An abnormal circumstance occurred in the No. 433 pegmatite: the pegmatite itself has very low Li contents (25 ppm to 92.3 ppm) but very high $\delta^7\text{Li}$ values (+5.4‰ to +7.6‰). In contrast, its adjacent country rocks have higher Li contents (894 to 1049 ppm).”

Line 119: please clearly state here that you are talking about dike 134 and the country rock

Thanks. This sentence has been replaced by “Lithium isotopic compositions of the country rocks adjacent to the No. 134 pegmatite are homogeneous ($\delta^7\text{Li} = +0.1\text{‰}$ to $+1.7\text{‰}$; Fig. 2d)” in the revised manuscript.

Line 150: compared to the

Thanks. It has been improved in the revised manuscript.

Line 233: evaporated till dry

Thanks. This problem has been modified in the revised manuscript.

Line 244: After evaporating till dry

Thanks. It has been improved in the revised manuscript.

Line 126-129: There are no references provided to support the statement that the Rb profile can be used to identify if fluids were involved. Additionally, the partitioning behaviour of Rb may vary compared to Li therefore the Rb profile might be shorter compared to the Li profile. Therefore, without any reference that support this statement this is not a valid argument to exclude the influence of fluids over longer distances (20 m). Please provide references and/or explain the involvement of fluids in more detail.

Thank you very much. In the revised manuscript, we have changed the sentence, as “These processes can be distinguished by comparing the profiles of Rb and Li in country rocks, as the transfer distance of Rb from pegmatite into country rocks is close to the infiltration distance of pegmatite-derived fluids³⁰.”. This point can be seen from the profiles in Shearer et al. (1986). Their study analyzed the country rock profiles around three pegmatites. The country rocks all contain volatile-rich minerals (such as muscovite, biotite, chlorite, tourmaline, etc.). Pegmatites are intruded in mid- to low-grade metamorphic rocks, which themselves contain volatile-rich minerals such as biotite, muscovite, and chlorite. There is great uncertainty in identifying whether these volatile-rich minerals are originally present in the country rocks (metamorphic rocks), or whether they are formed by metasomatism of pegmatite-derived fluids. However, the Bob Ingersoll No. 1 profile in this paper can clearly identify the scope of pegmatite-derived fluids, because the tourmaline in the country rock gradually decreases from the contact zone to the country rock (Fig. 3b in Shearer et al., 1986), and it disappears at 3-4m meters distance from the contact zone. Obviously, the tourmaline here corresponds to the

range of fluid infiltration. Then, it can be seen from the trace element changes in this profile (Fig. 6 in Shearer et al., 1986) that Rb is closer to the range of fluid infiltration, while Li is far beyond the range of fluid activity.

Shearer, C. K. et al. Pegmatite-wallrock interactions, Black Hills, South Dakota; interaction between pegmatite-derived fluids and quartz-mica schist wallrock. *Am. Mineral.* 71, 518–539 (1986).

Line 143: here it is stated that crystallisation in pegmatites is rapid – please add references, and further explain what you mean with rapid in this context. Do the authors mean with rapid a few days or decades. Please clarify.

Thanks. This sentence has been replaced by “Pegmatites are generally formed by crystallization under highly supercooling conditions^{36,37}, and the crystallization process is rapid, from several days^{8,33,34} to millennia³⁵.”.

Line 146: “although the Li content of the entire pegmatite dike is low at this time (Fig. 3).” I am not sure what the authors mean by that, is the Li concentration of the pegmatite low to start with? Please clarify.

Thanks. This sentence has been replaced by “although the average Li content of the entire pegmatite dike (including solidified parts, Li-enriched boundary layer, and residual melts) may be low.”

Line 298: why is -0.5 ‰ the average for the granites (average would be -1.25 ‰ when using the authors data), please clarify.

Thanks. This is a typing error. It is not the granites but the average value of the No. 134 pegmatite. The country rocks adjacent to the No. 134 pegmatite do not show a diffusion profile and thus experienced minimal kinetic fractionation. Therefore, the average $\delta^7\text{Li}$ value of the No. 134 pegmatite should be closest to the initial value of pegmatite melts. A sentence explaining this question has been added in the methods section, as “We assumed an initial $\delta^7\text{Li}$ value of -0.5‰ for the pegmatite melts that is the average value of the No. 134 pegmatite, because it experienced minimal kinetic fractionation and its average value should be closest to the initial value of pegmatite melts.”.

No analytical errors are provided for either the trace element measurements or the $\delta^7\text{Li}$ values. They are not presented in table 1, in the figures or the supplement file. Please provide the analytical errors on your measurements as these might affect the interpretation and/or your diffusion models.

Thanks very much. Analytical errors have been added in the methods part, as “The standard deviation based the long-term external repeatability of $\delta^7\text{Li}$ for the standard solution is approximately $\pm 0.52\%$ (2SD).” and “The relative standard deviation (RSD) is less than 5%.”. In addition, a sentence “The error bars are smaller than the symbol size because the standard solution of Li isotope analyses is approximately $\pm 0.52\%$ (2SD).” has been added in the captions of Fig. 2 and 3.

Major comments:

The introduction does not provide any information on previous work on Li diffusion in pegmatites or how their work is different compared to previous studies. I recommend to add a short summary of the findings of previous studies, e.g., what did they find and how did they explain it.

Thank you very much! A short summary has been added in the introduction this time, as “Although Li is a metallic element, its atomic mass is very small (6.941), only higher than H (1.008) and He (4.003). Together with its low ionic charge and size, Li has very high mobility. Therefore, Li can easily diffuse during high-temperature geological processes⁹. Diffusion process can be well recorded by Li isotopes. Lithium has two stable isotopes (i.e., ⁶Li and ⁷Li), and ⁶Li has a higher diffusion rate than ⁷Li (about 3% faster)^{10,11}. As a result, diffusion will form a unique Li isotope profile. Lithium isotope diffusion profile has been found around lithium pegmatites as well as granitic plutons^{12,13,14}, confirming that lithium diffusion occurs during the emplacement of pegmatites. However, the purpose of these studies was to reveal the kinetic fractionation behavior of lithium isotopes and did not pay attention to the effect of diffusion on Li content as well as Li mineralization of pegmatites^{12,13}. Moreover, pegmatites were assumed to be an infinite element Li reservoir in these studies^{12,13}, meaning that diffusion has no effect on Li content as well as mineralization ability of pegmatite itself. Based on an observation of substantially higher $\delta^7\text{Li}$ values of Li-poor pegmatites than Li-rich dikes, a recent study attributed the higher $\delta^7\text{Li}$ values of Li-poor pegmatites to diffusion-driven fractionation with a long duration, and speculated that diffusion may decrease the Li content of pegmatites¹⁵. Such a process of Li loss is critical for the formation of pegmatite Li deposit because it can determine the final grade and tonnage of pegmatite ore bodies. However, what factors account for the degree of Li loss by diffusion during pegmatite emplacement remains unclear.”

I recommend to add information about the samples, this could be provided in the supplementary material. For example, which minerals are present and to which percentage are they present? The mineral abundance will have an effect on the isotopic composition of the bulk sample. Also mention in the text how the samples were collected – in which distance from the contact. They are presented in table 1 but I would recommend to add an extra sentence. This could be added to the text around Line 108.

Thank you very much! More detailed descriptions have been added in the revised manuscript, as “Many pegmatite dikes in the Jiajika deposit have been numbered. For this study, we collected samples from three representative pegmatite dikes (Nos. 134, 308, and 433; Fig. 1b), the Majingzi granite, the surrounding country rocks, and the far-field country rocks. The No. 134 pegmatite is an extremely Li-rich dike and contains 0.28 million tonnes of Li, with an average Li_2O content of 1.38 wt.%²¹. Within the No. 134 pegmatite body, textures and minerals are variable, ranging from coarse to fine-grained textures. The main minerals include spodumene, quartz, muscovite, feldspar, tourmaline, etc. At a location where there is a clear contact boundary between the pegmatite and the country rock, profile samples of the country rock were collected in the horizontal direction perpendicular to the contact boundary (Table 1). The No. 308 pegmatite is also Li bearing but has a lower Li resource, with 0.0028 million tonnes of Li at an average grade of 0.5 wt.%²⁸. The internal texture and mineral assemblage of the No. 308 pegmatite are also variable. The main minerals are

quartz, spodumene, muscovite, feldspar, tourmaline, etc. Similarly, a country rock profile was selected for sampling (Table 1). The No. 433 dike is an Li-poor but Be-bearing pegmatite, and it is mainly composed of quartz, feldspar, muscovite, tourmaline, etc. The country rock surrounding the No. 433 dike was poorly exposed, and no suitable profile was found for sampling. Only two country rock samples were collected near the dike (<2m).”.

Throughout the main text it is not clearly stated what kind of diffusion the authors talk about. In line 132 to 133 they state that the fluid only worked over a short distance, it would help the reader and benefit the manuscript if the authors explained what kind of diffusion they are referring to either fluid- assisted grain-boundary diffusion or solid-state diffusion. It would help the manuscript if the authors provided arguments/explanations for why they prefer one over the other and how their data, e.g., the Rb profile support this statement.

Thanks very much. “...consistent with other fluid-assisted grain-boundary diffusion models^{12,13}. The fluid assisting diffusion here may be pegmatite-derived fluids (in areas close to the pegmatite), or it may be free water carried in the grain boundaries or cracks of low-grade metamorphic rocks (in areas farther away from the pegmatite).” has been added in the methods part. It should be explained here that the fluid infiltration mentioned here means that pegmatite-derived fluids flow along the fracture, interstice or other channels in country rocks, rather than diffusion. Fluid infiltration as well as reaction with host rocks is very common in magmatic-hydrothermal ore deposits and forms extensive hydrothermal alteration zones. The efficient diffusion of Li in schist is fluid-assisted grain-boundary diffusion (Teng et al., 2006; Liu et al., 2010), while solid-state volume diffusion is quite inefficient, several orders of magnitude lower than grain-boundary diffusion (Zhang, 2010). The fluid assisting diffusion here may be pegmatite-derived fluids (in areas close to the pegmatite), or it may be free water carried in the grain boundaries or cracks of low-grade metamorphic rocks (in areas farther away from the pegmatite).

Zhang, Y. (2010). Diffusion in minerals and melts: theoretical background. *Reviews in mineralogy and geochemistry*, 72(1), 5-59.

In line 136 to 137 the authors explain that their data confirms that Li diffuses from the pegmatite into the country rock. The way this sentence is phrased it makes it sound like this is a new finding, however, the study by Teng et al. 2006 already showed this, please add a reference here. Further, it is not clear why only dike 308 was modelled or why the dike was chosen for the model. I assume it is based on the concentration and isotope profiles, please clarify this in the text.

Thank you very much. In the revised manuscript, we have modified this sentence, as “It is also supported by the Li isotope profile (Fig. 3d), i.e., continuously decreases in Li content and the large fractionation of Li isotope confirm that it is a diffusion-induced profile¹². On the basis of the Li diffusion modeling for the No. 308 pegmatite, the measured Li content and Li isotope profiles are consistent with the simulated profile at 20 years after emplacement (Fig. 4).”.

It should be explained that we are not to say that lithium diffusion around pegmatites is a new discovery of this study. Here we attempt to demonstrate that the composition gradient is caused by diffusion, rather than fluid infiltration. Teng et al. (2006) first reported Li diffusion around the Tin Mountain pegmatite using Li isotopes. But in other pegmatite deposits, if we want to explore how

this diffusion affects Li mineralization, we think we should demonstrate the presence of lithium diffusion in the studied pegmatite deposit before discussion.

The authors show that their concentration and isotope profile must have formed within 35 years and state that this confirms that diffusion must have driven this migration of Li. This does not confirm Li diffusion; the authors are using a circular argument here. The fact that it continuously decreases in concentration and the large fractionation of the isotopes confirm that it is a diffusion profile. Please rephrase this part of the text. State clearly that due to this profile a diffusion model is applied to the data, this diffusion model assumes that the Li concentration and $\delta^{7}\text{Li}$ profile must have formed within 35 years. If the authors mean that their result of 35 years confirms that it is diffusion this needs to be explained.

Thank you very much. In the calculation of the diffusion time (No. 308 dike) in the first draft, we made a small mistake. The width of the dike used in the calculation was half of the real width. The width of the No. 308 dike is 30m. It was divided by 2 twice (maybe because Excel was opened at different times, and I forgot that I had divided it once), resulting in the width used in the initial calculation being 7.5m, which should actually be 15m. In the revised manuscript, we have corrected this problem, resulting in a diffusion time of 20 years.

The problem you mentioned has been improved in the revised manuscript, as "It is also supported by the Li isotope profile (Fig. 3d), i.e., continuously decreases in Li content and the large

fractionation of Li isotope confirm that it is a diffusion-induced profile¹². On the basis of the Li diffusion modeling for the No. 308 pegmatite, the measured Li content and Li isotope profiles are consistent with the simulated profile at 20 years after emplacement (Fig. 4).’.

Additionally, I think it would improve the paper if the authors could put their result in context with others papers that focus on cooling rates of pegmatites, e.g., Simmons et al. (2008), Webber et al., (1999), Phelps et al. (2020). Cooling rates of pegmatites are a strongly debated topic and the findings of the authors could add valuable information to the discussion.

Thank you very much for sharing the idea and references! The following discussion about this topic has been added in the revised manuscript, as “The cooling time of pegmatite dikes obtained by different methods is variable, ranging from several days^{8,33,34} to millenniums³⁵. Our result show that the cooling time of the No. 308 pegmatite in Jiajika is ~20 years. These different cooling times may all be reasonable within a pegmatite deposit. Because the conditions and intrusion environments of each dike are different, and the cooling rates vary greatly.”.

Line 138 to 150 and Figure 3. Based on the data presented by the authors I do not understand how the authors arrive at the conclusion that the pegmatite lost all Li to the country rock resulting in an enrichment of the country rock and a full depletion of the pegmatite. Do the authors assume the melt was already low and all the Li is leached from the dike? Several measurements of the Schist that is related to dike 134 have similarly high concentrations in Li. I would recommend to find more supportive arguments for why Li should diffuse completely out of the dike into the country rock, what is the difference between the dikes or the country rock that this would happen? Are other minerals present in the country rock around this dike compared to dike 308 that Li preferentially partitions into the host rock? Are you assuming all Li is leached from the pegmatite into the country rock by fluids? This part of the text needs to be expanded and explained in more detail.

Thank you very much! Maybe our expressions are not clear. The premise of diffusion is the existence of a concentration gradient. Regardless of whether the initial melt of the No. 433 pegmatite is Li-rich or Li-poor, crystallization under high undercooling will form a boundary layer with a relatively high Li content than the original pegmatite melt (London, 2018). The Li content in the boundary layer is likely higher than that in the country rock, so Li can diffuse from the boundary layer into the country rock. Although the Li content in the boundary layer is relatively high, the average Li content of the entire dike (including solidified parts, Li-enriched boundary layer, and residual melts) is still low. When the pegmatite dike is completely solidified, i.e., the boundary layer and residual melt disappear, it will result in a lower Li content in the pegmatite dike. This inferred process is mainly based on Li content and Li isotope characteristics, and we think it is reasonable. However, it is difficult for us to provide other evidence in this study. Given your suggestions, we have changed the expressions, as “Interestingly, the No. 433 pegmatite has very low Li contents (25.0–92.3 ppm Li; Table 1), but its adjacent country rocks have very high Li contents (894–1049 ppm Li; Table 1). There may be multiple explanations for this pattern. For example, the No. 433 dike is originally Li-poor and has a very high $\delta^7\text{Li}$ value. Its country rocks may originally have a high Li content, or they were contaminated by other Li-rich pegmatites through diffusion. Here, we provide another possible scenario that can explain Li elemental and isotopic characteristics of the No. 433 dike and its country

rocks together. Pegmatites are generally formed by crystallization under highly supercooling conditions^{36,37}, and the crystallization process is rapid, from several days^{8,33,34} to millenniums³⁵. As a consequence, an Li-enriched boundary layer of melt adjacent to crystal surfaces forms during the solidification of pegmatite⁸. Lithium migration from an Li-enriched boundary layer into country rocks occurs continuously, although the average Li content of the entire pegmatite dike (including solidified parts, Li-enriched boundary layer, and residual melts) may be low. The diffusion process continues until the entire pegmatite body is completely consolidated. Finally, the entire pegmatite has lost most of its Li by diffusion, resulting in a scenario where the pegmatite is Li poor but with high $\delta^7\text{Li}$ value, and the country rock has a higher Li content compared to the final Li content of the pegmatite dike.”.

It is not clear if the blue line in figure 3 is supposed to be the diffusion profile for the $\delta^7\text{Li}$ values, please add this information to the figure caption. If this is the diffusion profile for the $\delta^7\text{Li}$ values, how did the authors model this profile? Is it based on figure 2h? If so why would this be valid, since both dikes 433 and 308 behave differently. Throughout the text it is also not explained why these dikes are so different and how one dike lost all its Li whereas the other did not, it would greatly improve the manuscript if the authors further expanded on this.

Thank you very much. The diffusion curves in Figure 4 (in the revised version) were calculated using diffusion equation. The blue lines in Figure 8 (in the revised version) are not a quantitative result of diffusion, but an approximate illustration of the effect of diffusion on Li content changes. In the revised manuscript, we provide a more comprehensive figure and describe it in detail in the figure caption, as:

Fig. 8 The diffusion loss of Li when pegmatites intrude into country rocks with different

temperatures. Within the country rocks with different temperatures, we considered two situations where the initial pegmatite melt was Li-rich and Li-poor. In low-temperature country rocks, pegmatites cool very quickly, and the timescale for effective diffusion of Li is very short. Whether initial pegmatite melts are Li-rich or Li-poor, the amount of Li loss is very limited. In high-temperature country rocks, pegmatites cool very slowly, and it takes a very long timescale for Li diffusion. Even if the initial pegmatite melt is Li-rich, most of Li will migrate into the country rocks through diffusion. In the Jiajika deposit, the No. 134 pegmatite corresponds to the scenario of **b**; the No. 308 pegmatite corresponds to the process of **d**; the No. 433 pegmatite may correspond to the scenario of **e** or **f**. The blue lines are not a quantitative result of diffusion modeling, but an approximate illustration of the effect of diffusion on Li content changes.

Figure 2 – I believe this figure could be improved by 1) giving a, b and c the same scale so they are comparable and give d, e and f the same scale; 2) I would combine b and c with each other and e and f with each other and instead show the data for the third dike in c and f; 3) I believe 2g and 2h should be a separate figure. In the figure caption I would also mention that for the Li concentrations of the dikes the average values are used or the measured data could be plotted in that figure as well. It is not clear and also not explained why the authors model for 10, 50 and 100 years and why in 2g and 2h no model for 35 years is shown. Please give further information in the text. It is also not clear why the authors use 3500 ppm in 2g) as the initial concentration of pegmatite 308 since the concentration of this dike was measured, why is literature data used here? Similarly, in 2c) the boundary conditions that are applied by the authors are not clearly explained and it is not explained why these boundary conditions are applied. Same is true for the isotope models. Please clarify in the text. How would different diffusion coefficients influence the results and why was this specific diffusion coefficient applied? The partitioning of Li depends on the minerals that are present in the host rock and in the pegmatite however this issue is not addressed at all in the text or the methods, please explain further in the text.

Thank you very much. In the revised manuscript, this figure has been separated into two figures (Figure 3 and Figure 4 in the revised manuscript), and the two figures were also modified according to your suggestions.

More descriptions have been added in response to the above questions, as “Although this study analyzed the Li content of different pegmatites in Jiajika, the sample data are limited and their Li contents are extremely heterogeneous. It is uncertain that the average value of these samples represents the true Li content of the pegmatite dikes. In order to make our approach of the broad significance, here we use the statistical results provided by the previous study⁵⁰, i.e., Li-rich pegmatites, such as the No. 134 pegmatite in Jiajika, generally contain ~7000 ppm Li; zoned pegmatites, such as the No. 308 pegmatite in Jiajika, generally contain ~3500 ppm Li. First, we calculated the loss of Li in pegmatites only from the perspective of Li diffusion (Fig. 6). According to the diffusion equation, two key factors affecting the diffusion loss of Li in pegmatite dikes are diffusion timescale and dike width. Lithium diffusion from pegmatites into country rocks occurs only during the cooling of pegmatites and ceases when the pegmatites are completely crystallized. Therefore, the timescale of diffusion is also the cooling time of pegmatite dikes. Diffusion modeling of a Li-rich pegmatite (initial lithium content = 7000 ppm) show that its Li content in the pegmatite dike will decrease with increasing cooling time. The wider the pegmatite dike, the rate of Li loss is

slower (Figure 6). For zoned pegmatites, such as the No. 308 pegmatite in Jiajika, if it has an initial Li content of 3500 ppm, its Li content decreases to less than 2000 ppm after several decades of diffusion (Fig. 4a). In such a case, the pegmatite dike loses its ore-forming potential, as supported by the low grade and tonnage of the No. 308 pegmatite.”

More details about diffusion modeling are present in the methods part. In particular, for the diffusion coefficient, we cited a previous result that is fluid-assisted grain boundary diffusion in schist (10^{-7} m²/s; Teng et al., 2006; Liu et al., 2010). For the β value, we obtained it based on repeated attempts based on the Li content and Li isotope profile of the No. 308 pegmatite. Only when the β value = 0.08, the measured Li content and Li isotope profiles can be simulated concurrently.

In the section “Country-rock temperature: a key control on Li pegmatite genesis” the authors refer to their figure 4 and 5, however the authors do not explain their figures. It is not explained why these temperatures for the country rock were chosen, why the different widths for the dikes were chosen. It would improve the manuscript if the authors explained their figures in more detail and how they chose the values for the parameters. How does the model change from 4b to 4h? What can we learn from that? How does it affect the cooling time? Why was 750 °C used as the starting temperature, how would other starting temperatures affect this model?

Thank you very much! In the revised manuscript, we have added a new part in the discussion, and all of above questions have been answered in this part, as:

“Timescale of pegmatite cooling

Obviously, the amount of Li migration from the pegmatite dike into country rocks depends on the duration of the cooling of pegmatites. The cooling time of pegmatite dike can be constrained by thermal modeling^{8,35}. Here, a conductive cooling model (HEAT3D³⁸) was employed to track the thermal history of pegmatite dikes after their emplacement. Because pegmatites do not contain inherited phenocrysts⁸, the initial pegmatite melts should approach the liquidus temperature of high-silica melts. The Macusani obsidian was considered to be a composition close to lithium pegmatite^{36,39}. Its liquidus temperature is close to $\sim 750^{\circ}\text{C}$ ³⁹ at 200 MPa (the average pressure of pegmatite emplacement⁸) with 6 wt% H₂O content (the saturated H₂O content of rhyolitic melts at 200 MPa⁴⁰). Therefore, we set the initial temperature of the pegmatite melt to 750 °C. Parameters for the pegmatite-forming melt are bulk density of 2300 kg/m³, heat capacity of 1100 J/ kg*K, and thermal conductivity of 1.5 W/m*K^{8,41}.

The country rocks in the Jiajika deposit are schists and metamorphic sandstones, which has a bulk density of 2790 kg/m³, specific heat of 1100 J/kg K⁴². Thermal conductivity of schists and metamorphic sandstones was obtained by $\lambda(200^{\circ}\text{C}) = 0.75 + 705/(350 + 200)^{42}$. Pegmatite dikes were emplaced within the thermal aureole heated by their parent pluton⁴³. Therefore, the country rock near the parent pluton has a high temperature, and the country rock far away from the pluton is close to the normal geothermal gradient. The crystallization temperature of many minerals in pegmatite is as low as $\sim 375\text{--}475^{\circ}\text{C}$ ⁴⁴, so 400 °C was chosen as the upper limit of the temperature of country rocks. Under normal geothermal gradient (i.e., $\sim 20^{\circ}\text{C}/\text{km}$), the rock temperature corresponding to 200 MPa is $\sim 130^{\circ}\text{C}$, so 100 °C was set as the lower limit of the temperature of country rocks. Most pegmatite dikes in nature are small, with widths of a few meters⁴⁵. A very wide one is the Tanco pegmatites that have an average thickness of $\sim 40\text{m}$ ⁸. In the thermal modeling, we therefore set the minimum and maximum values of the pegmatite width to be 2 m and 40 m (Fig.

5), respectively.

A representative thermal profile is shown in Figure 5a. We simulated the intrusion of pegmatites with different widths into country rocks with different temperatures. Figures 5b–5h correspond to different country rocks with variable temperatures, respectively. It shows that when the country rock temperature is the same, thin pegmatite dikes have a shorter cooling time than wide dikes (Fig. 5i). When the dike width is the same, the higher the temperature of the country rock, the longer the cooling time of the pegmatite dikes. These observations are generally consistent with previous results³³. Two factors influence the cooling time of pegmatite dikes: the temperature of country rocks at the time of pegmatite emplacement and the width of pegmatite dikes (Fig. 5i). Wider pegmatite dikes with higher country-rock temperature have longer cooling times compared with thinner pegmatite dikes with lower country-rock temperature. In addition, the initial temperature of pegmatite melt can also affect the cooling time of pegmatite dikes, with higher temperature melts requiring longer cooling times. However, this factor is more complicated. Because the initial temperature of pegmatite melt is affected by melt composition, volatile content, and the content of fluxing components (such as Li and B)⁴⁶, and it needs to be considered in conjunction with phase equilibrium. This study set a same temperature of the initial pegmatite melt, but this does not affect our discussion of the influence of dike width and country-rock temperature, because the effects of these factors on pegmatite cooling time are independent of each other.”

In the text it says 3500 ppm was chosen as the boundary condition for this model, why and also in Figure 5a the boundary condition is more around 7000 ppm? Are the authors modelling different dikes? This is not clear from the figure caption or the main text, please clarify in the text. The authors state that the width of the dikes places a role for the cooling time, this has been addressed by others studies before, please reference them and also how do the findings in this study compare to other studies? Why are the concentrations in Figure 5 different to Figure 2? Figure 5 is not sufficiently explained in the text, please give more information. In the thermal modelling section, the authors could give more explanation for the values they use for their models.

Thank you very much! In the revised manuscript, we have added more descriptions in the section “Country-rock temperature: a key control on Li pegmatite genesis”, and all of above questions have been answered in this part, as:

“Although this study analyzed the Li content of different pegmatites in Jiajika, the sample data are limited and their Li contents are extremely heterogeneous. It is uncertain that the average value of these samples represents the true Li content of the pegmatite dikes. In order to make our approach of the broad significance, here we use the statistical results provided by the previous study⁵⁰, i.e., Li-rich pegmatites, such as the No. 134 pegmatite in Jiajika, generally contain ~7000 ppm Li; zoned pegmatites, such as the No. 308 pegmatite in Jiajika, generally contain ~3500 ppm Li. First, we calculated the loss of Li in pegmatites only from the perspective of Li diffusion (Fig. 6). According to the diffusion equation, two key factors affecting the diffusion loss of Li in pegmatite dikes are diffusion timescale and dike width. Lithium diffusion from pegmatites into country rocks occurs only during the cooling of pegmatites and ceases when the pegmatites are completely crystallized. Therefore, the timescale of diffusion is also the cooling time of pegmatite dikes. Diffusion modeling of a Li-rich pegmatite (initial lithium content = 7000 ppm) show that its Li content in the pegmatite dike will decrease with increasing cooling time. The wider the pegmatite dike, the rate of Li loss is

slower (Figure 6). For zoned pegmatites, such as the No. 308 pegmatite in Jiajika, if it has an initial Li content of 3500 ppm, its Li content decreases to less than 2000 ppm after several decades of diffusion (Fig. 4a). In such a case, the pegmatite dike loses its ore-forming potential, as supported by the low grade and tonnage of the No. 308 pegmatite.

Then, we input the cooling time (i.e., the results of the cooling time of pegmatite dikes with different widths and country-rock temperatures) obtained from the thermal modeling, into the Li diffusion equation, leading to the output of the final Li contents and $\delta^7\text{Li}$ values of pegmatite dikes after diffusion (Fig. 7). There is an unexpected result: the final Li content of pegmatite dikes depends mainly on the temperature of country rocks, whereas the effect of the width of pegmatite dikes is negligible (Fig. 7a and 7c). Although the loss of Li by diffusion from a wider dike is slower (Fig. 6), its heat loss is also slower (Fig. 5), resulting in a longer cooling time (Fig. 5i). As a consequence, the ratio of Li loss to initial Li content for wide dikes is not much different from that of thin dikes (Fig. 7a).”

Response to Reviewer #2

Reviewer #2 (Remarks to the Author):

We have reviewed this manuscript entitled: “Pegmatite lithium deposits formed within low-temperature 1 country rocks” written by Zhou et al. The supply of Li for electric vehicles has a great risk in the next years, and thus it is critical to decode the nature of pegmatite and their implications for exploration. In this study, the authors present a detailed Li isotopes from three distinct pegmatites at Jiajika to provide the new constraints on hard-rock type Li deposits. They suggest a novel interpretation that the low temperature country rock is the crucial parameter for rapid ore formation. This manuscript is well organized. However, some parts need further modification before publications. I recommend a major revision on it.

We are very grateful to Reviewer #2 very much for providing constructive, insightful and helpful reviews of this manuscript! These comments greatly improved the manuscript. Thanks for your time and patience!

General concerns:

The authors suggest that the country-rock profile line of the No. 308 pegmatite can be well explained by the diffusion model of Li via the variations of Rb-Li contents and Li isotopes. However, the trends of the 134 pegmatite is inconsistent with model. The authors stress that the gradually increasing Li content on the right should be affected by another pegmatite dike, but it is not convinced for me. From the geologic map, the size of the 134 pegmatite is clearly bigger than the nearby dikes, and thus, it is hard to believe that the unknown little sized dike shows stronger effects of diffusion.

Thank you very much! Given your comments, we provide a more comprehensive figure concerning the diffusion processes in the revised manuscript.

Fig. 8 The diffusion loss of Li when pegmatites intrude into country rocks with different temperatures. Within the country rocks with different temperatures, we considered two situations where the initial pegmatite melt was Li-rich and Li-poor. In low-temperature country rocks, pegmatites cool very quickly, and the timescale for effective diffusion of Li is very short. Whether initial pegmatite melts are Li-rich or Li-poor, the amount of Li loss is very limited. In high-temperature country rocks, pegmatites cool very slowly, and it takes a very long timescale for Li diffusion. Even if the initial pegmatite melt is Li-rich, most of Li will migrate into the country rocks through diffusion. In the Jiajika deposit, the No. 134 pegmatite corresponds to the scenario of **b**; the No. 308 pegmatite corresponds to the process of **d**; the No. 433 pegmatite may correspond to the scenario of **e** or **f**. The blue lines are not a quantitative result of diffusion modeling, but an approximate illustration of the effect of diffusion on Li content changes.

For the No. 134 pegmatite, at a location where there is a clear contact boundary between the pegmatite and the country rock, profile samples of the country rock were collected in the horizontal direction perpendicular to the contact boundary. In the field, this pegmatite is very wide, but it is uncertain whether the entire No. 134 pegmatite is a single dike or an assemblage of many small pegmatite dikes, because we have observed that some dikes are superimposed on other dikes in some places. Therefore, the other dike mentioned in the text does not necessarily correspond to the small dike shown in Fig. 1b.

The Li content of the country rocks near the sampled dike within the No. 134 pegmatite is very low (the three points on the left in Fig. 3a), indicating very limited transfer of Li from the pegmatite into the adjacent country rocks. The gradually increasing trend starting from the fourth point should be affected by another pegmatite dike. If it was affected by the sampled dike, there should be a gradually decreasing trend.

This paper emphasizes that temperature-controlled lithium diffusion in surrounding rocks affects the lithium mineralization of pegmatites, so what are the conditions for lithium diffusion? and why is there almost no diffusion effect in 134 veins in this study.

Thank you very much. Maybe we didn't express it clearly in the first draft. In the revised manuscript, we have added a more comprehensive figure about the diffusion processes (Fig. 8). In fact, any pegmatite has diffusion effect, including the No. 134 pegmatite. But the duration of the diffusion process in different dikes is variable. Among the three pegmatite dikes we analyzed, the No. 134 pegmatite has a shortest timescale of Li diffusion, resulting in negligible Li loss of this pegmatite. The diffusion of the No. 308 pegmatite lasted for ~20 years, causing a part of the Li (~17%) in the pegmatite to migrate into the country rock. The No. 433 pegmatite may have the strongest diffusion effect, causing most of its lithium to diffuse into the country rock. In a pegmatite deposit, different dikes have different degrees of diffusion effect. We think a key factor controlling this difference is the temperature of country rocks, which is a new discovery of this study and has never been mentioned before (as far as we know).

The diffusion for the Li-poor 433 pegmatite vein is unreliable. Firstly, following the fractionation model, the Li-poor pegmatites formed in the early stage, and they are generally featured by low concentrations of Li. It does not need the diffusion model to explain the low Li in this pegmatite. Moreover, the high concentrations of wall rocks have been reported, and the diffusion model is

unnecessary for this phenomenon. Thirdly, if the diffusion occurred in 433 pegmatite vein, but why it did not occur in Li-rich 134 and 308 veins. Lastly, the data of NO.433 vein should also be given in the paper.

Thank you very much. Maybe our expressions were not clear. First of all, we did not say that the initial melt of the No. 433 pegmatite was necessarily Li-rich. It might be Li-poor, or it might have had a higher Li content than it does now. Here, we introduce diffusion to explain it because it has abnormally high $\delta^7\text{Li}$ values, but its country rocks have higher Li content. Given your comments, we have modified the expressions, as “Interestingly, the No. 433 pegmatite has very low Li contents (25.0–92.3 ppm Li; Table 1), but its adjacent country rocks have very high Li contents (894–1049 ppm Li; Table 1). There may be multiple explanations for this pattern. For example, the No. 433 dike is originally Li-poor and has a very high $\delta^7\text{Li}$ value. Its country rocks may originally have a high Li content, or they were contaminated by other Li-rich pegmatites through diffusion. Here, we provide another possible scenario that can explain Li elemental and isotopic characteristics of the No. 433 dike and its country rocks together. Pegmatites are generally formed by crystallization under highly supercooling conditions^{36,37}, and the crystallization process is rapid, from several days^{8,33,34} to millenniums³⁵. As a consequence, an Li-enriched boundary layer of melt adjacent to crystal surfaces forms during the solidification of pegmatite⁸. Lithium migration from an Li-enriched boundary layer into country rocks occurs continuously, although the average Li content of the entire pegmatite dike (including solidified parts, Li-enriched boundary layer, and residual melts) may be low. The diffusion process continues until the entire pegmatite body is completely consolidated. Finally, the entire pegmatite has lost most of its Li by diffusion, resulting in a scenario where the pegmatite is Li poor but with high $\delta^7\text{Li}$ value, and the country rock has a higher Li content compared to the final Li content of the pegmatite dike.”.

In addition, the data of the No.433 dike has been added in the result part in the revised version, as “An abnormal circumstance occurred in the No. 433 pegmatite: the pegmatite itself has very low Li contents (25 ppm to 92.3 ppm) but very high $\delta^7\text{Li}$ values (+5.4‰ to +7.6‰). In contrast, its adjacent country rocks have higher Li contents (894 to 1049 ppm).”

In this contribution, the authors emphasized the critical role of the temperatures of wall rocks in the formation of pegmatite Li deposit. However, the authors do not give any data of the temperature of wall rocks at the Jiajika deposit when the pegmatite emplaced. The depth of these wall rocks is related to their temperatures considering regions geothermal gradient. So what is the emplacement depth may provide the clues for temperature of wall rocks.

Thank you very much. In the Jiajika deposit, estimates using fluid inclusion within quartz crystals indicate that the pressure at which the pegmatites formed varied from ~150 MPa to ~420 MPa (Wang et al., 2023). Such a pressure fluctuation is not necessarily the true difference in formation depth, because there is considerable uncertainty in the pressure estimation using fluid inclusions.

The temperature difference in country rocks emphasized in this study is not caused by the geothermal gradient, but by the thermal field generated by the parent pluton. Areas close to the parent pluton have high temperatures, while locations far away from the pluton have low temperatures. It can be evidenced by the thermal metamorphism zoning in Jiajika (Fig. 1b). In the revised manuscript, we have added a figure of the metamorphism zonation and added a description

of related content in the text, as “As evidenced by the thermal metamorphism around the Majingzi pluton that from the inner to outer aureoles, it changes from a high-temperature staurolite zone to a low-temperature biotite zone (Fig. 1b).”

Wang, G. G. et al. Fluid properties and ore-forming process of the giant Jiajika pegmatite Li deposit, western China. *Ore Geol. Rev.* 160, 105613 (2023).

The authors considered the temperature of wall rocks as main contributor to the cooling time of pegmatite dikes. But to the best of our knowledge, pegmatites were formed under supercooling conditions, which means that the composition of the pegmatitic melts may be influence the cooling time. Please explain the validness of your theoretical model in the geologic conditions.

Thank you very much! Yes, now there is a consensus that pegmatites crystallized under high supercooling conditions. Under high supercooling conditions, crystals have a high growth rate, resulting in crystals can grow to a large grain in a short timescale. There are two factors that lead to high supercooling. One is rapid cooling, and the other is an increase in the liquidus temperature of magmas. This manuscript emphasizes the effect of rapid cooling, and we believe it is consistent with geological conditions. As for the second case, that is, how the composition of the pegmatite melt affects the cooling process. This factor is also very important, especially the volatile components and flux components in pegmatite melts, which have a significant influence on the cooling of pegmatite melts. Given your comments, we have added more discussions about this question in the revised manuscript, as “In addition, the initial temperature of pegmatite melt can also affect the cooling time of pegmatite dikes, with higher temperature melts requiring longer cooling times. However, this factor is more complicated. Because the initial temperature of pegmatite melt is affected by melt composition, volatile content, and the content of fluxing components (such as Li and B)⁴⁶, and it needs to be considered in conjunction with phase equilibrium. This study set a same temperature of the initial pegmatite melt, but this does not affect our discussion of the influence of dike width and country-rock temperature, because the effects of these factors on pegmatite cooling time are independent of each other.”.

The mechanism of lithium isotope fractionation is not clear. Are there any other mechanisms (e.g. source heterogeneity, supercritical fluid) besides fluid metasomatism and diffusion to make Li isotope variation?

Thanks very much. Yes, source inhomogeneity may lead to different lithium isotope compositions. However, in the Jiajika deposit, almost all of the studies support that the pegmatites were derived from the Majingzi pluton. The lithium isotope variation of this pluton is -3.1% to $+1.9\%$ (Fig. 2), and the country rocks near the No. 308 pegmatite have a lithium isotope change of -13.8% to $+1.5\%$. Apparently, the Li isotope differences in the parent pluton are not sufficient to produce the large Li isotope changes of the country rocks. The influence of supercritical fluid is also possible. Teng et al. (2006, AM) discussed this question and found that the lithium isotope fractionation during fluid activity was $<2\%$. In general, continuously decreases in Li content and the large fractionation of Li isotope confirm that it is a diffusion-induced profile.

Teng, F. Z. et al. Lithium isotopic systematics of granites and pegmatites from the Black Hills, South Dakota. *Am. Miner.* 91, 1488–1498 (2006b).

The natural case of the Jiajika deposit is good. Can you give more specific natural cases to support your idea that cooling time can severely affect the formation of the Li deposit? If it is a common phenomenon for rare metal pegmatite, then the new model would be important.

Thank you very much. In the manuscript, we have tried our best to find more natural examples and cited other cases from the perspective of Li isotopes, as “Another piece of evidence supporting our conclusion is the contrasting Li isotopic compositions of Li-poor and Li-rich pegmatites. In the Jiajika deposit, Li-rich pegmatites have lower $\delta^7\text{Li}$ values compared with Li-poor pegmatites, and Li-poor pegmatites show wide variations in Li isotopic composition and many samples have high

$\delta^7\text{Li}$ values (up to +7.6‰; Fig. 2). This pattern is consistent with results from thermal and diffusion modeling, which show that when pegmatite dikes intrude high-temperature country rocks, they lose most of their Li and have elevated $\delta^7\text{Li}$ values (Fig. 7c and 7d). A similar pattern (i.e., pegmatite samples with elevated $\delta^7\text{Li}$ are derived mainly from Li-poor dikes) can also be seen in other pegmatite districts^{15,55,56,67}.”.

Jiajika is a very typical and noteworthy lithium pegmatite, probably the largest pegmatite Li deposit in the world. We believe the conclusion of this study should be representative.

The cooling time of pegmatite dikes is very fast, and it is difficult to use traditional geochronological methods to constrain it now. This manuscript discussed this topic from the perspective of Li isotopes. We think that what may be more effective to solve this difficulty is the diffusion chronology (Costa et al., 2020). This method can reach timing units of "years", "days" and even "hours". We are preparing the work about the cooling time of pegmatite dikes using diffusion chronology.

Costa, F., Shea, T., & Ubide, T. (2020). Diffusion chronometry and the timescales of magmatic processes. *Nature Reviews Earth & Environment*, 1(4), 201-214.

Specific concerns:

The granite name is “Majingzi” not “Majinzi”. Please check the text, tables, and figures.

Thank you very much! This problem has been improved.

Line 119, please clarify that lithium isotopic compositions of the No. 134 pegmatite are homogeneous.

Thank you very much! This sentence has been replaced by “Lithium isotopic compositions of the country rocks adjacent to the No. 134 pegmatite are homogeneous ($\delta^7\text{Li} = +0.1\text{‰}$ to $+1.7\text{‰}$; Fig. 3c).”.

REVIEWERS' COMMENTS

Reviewer #1 (Remarks to the Author):

I appreciate the work the authors have put in to improve the manuscript.

Reviewer #2 (Remarks to the Author):

Review: Manuscript Number: NCOMMS-23-55961

Title: Pegmatite lithium deposits formed within low-temperature country rocks

The authors have carefully treated the suggestions, and the clarity and quality of the revised manuscript has been significantly improved. In this manuscript, a commonly overlooked, but potentially essential, factor in the genesis of pegmatite lithium deposits has been highlighted, which is a great contribution to this field of researches. With that being said, we still have some reservations regarding the robustness of the conclusion. Below, we give our major concerns that are based on geological considerations. We think that these concerns should be appropriately addressed before publication, and accordingly, a moderate revision is recommended.

General comments:

λ In Fig. 3, two contrasting Li isotope and concentration trends are exhibited from pegmatite dikes outwards to wall rocks. For the trend around No. 134 pegmatite, the authors stressed that it is influenced by a nearby pegmatite dike. Please provide a photograph that clearly displays the spatial relationship between No. 134 and its nearby pegmatite dikes.

λ In lines 208-220, the authors explained how the diffusion of Li can lead to the observed signatures of Li isotopes and contents of No. 433 pegmatite dike and its country rocks. However, this interpretation is quite confusing as the Li contents of surrounding wall rocks (894-1049 ppm) is distinctly higher than those of No. 433 dike (25.0-92.3 ppm). As diffusion is driven by differences in element concentrations, it is hard to envisage that, if Li is diffused from pegmatite dike to wall rocks, the final Li content of wall rocks would be higher than the pegmatite dikes. Although the boundary layer theory was invoked in their discussion, we don't think the distinctly low Li contents in pegmatites can be explained as the result of the diffusion process.

λ In line 230, the authors assume that the initial temperature of the pegmatite melt is 750°C, which is debatable. Several related information can be shown below: 1) It is estimated that Li-mineralized pegmatites have a mean crystallization temperature of $\sim 510 \pm 25$ °C compared to Be- and rare earth element (REE)-mineralized pegmatites with mean temperatures of $\sim 550 \pm 45$ °C (McCaffrey et al., 2023); 2) Granites, as the source of pegmatite, crystallized at about 700°C and could undergo overcooling to form pegmatites (London et al., 2018). So, the initial temperature of pegmatite melt should be modified.

London, D., 2018. Ore-forming processes within granitic pegmatites. *Ore Geology Reviews*, 101: 349-383.
McCaffrey, D.M. and Jowitt, S.M., 2023. The crystallization temperature of granitic pegmatites: The important relationship between undercooling and critical metal prospectivity. *Earth-Science Reviews*, 244: 104541.

λ In Fig.1c, the authors think that the age of granite intrusion and regional metamorphism are consistent, which is debatable. Two main stages of regional metamorphism and deformation are documented in Jiajika. The regional metamorphism and granite intrusion are related to early-stage Barrovian metamorphism and late-stage Buchan metamorphism, respectively (Zhu et al.,2024). Here should be modified.

Zhu, J., Zhu, W., Xu, Z., Zheng, B., Li, G., Lin, H. and Gao, J., 2024. The composite metamorphic sequence in the Jiajika gneiss dome, Songpan-Ganze orogenic belt, eastern Tibet: P-T-D-t evolution and implications for lithium mineralization. GSA Bulletin. <https://doi.org/10.1130/B37747.1>

Specific comments:

λ In Fig.1, the “Majinzi” should be “Majingzi”;

λ In line 118, the “depoists” should be “deposits”.

Review: Manuscript Number: NCOMMS-23-55961

Title: Pegmatite lithium deposits formed within low-temperature country rocks

I have reviewed the revised manuscript by Zhou et al. titled 'Pegmatite lithium deposits formed within low-temperature country rocks' and their replies to the reviewers. The authors have addressed all concerns of the reviewers, and the manuscript has greatly improved. I still believe this paper will be a great contribution to our knowledge and is well suited for Nature Communications. I therefore recommend publishing the paper, after very minor changes.

I have included some suggestions for rewording in the attached file.

Response to Reviewer #1

Reviewer #1 (Remarks to the Author):

I appreciate the work the authors have put in to improve the manuscript.

I have reviewed the revised manuscript by Zhou et al. titled 'Pegmatite lithium deposits formed within low-temperature country rocks' and their replies to the reviewers. The authors have addressed all concerns of the reviewers, and the manuscript has greatly improved. I still believe this paper will be a great contribution to our knowledge and is well suited for Nature Communications. I therefore recommend publishing the paper, after very minor changes.

We are very grateful to Reviewer #1 for providing constructive and detailed reviews of this manuscript! The revised manuscript had been corrected according to the comments in the attached file. Thanks for your time and patience!

Changes including:

Line 24 "lead" has been replaced by "lead to".

Line 27 "economically relevant" has been added.

Line 31 "a surprising result" has been removed.

Line 33 "the surrounding" has been added.

Line 72 "form a" has been replaced by "lead to".

Line 156 "In contrast" has been replaced by "In contrast to the pegmatite".

Line 196 "On the basis of" has been replaced by "Based on".

Line 274 "In contrast, this study explores" has been replaced by "This study, however, explores"

Line 284 "In order to make our approach of the broad significance, here we" has been replaced by
"To make our approach of broader significance, we"

Line 296 "The wider the pegmatite dike, the rate of Li loss is slower (Figure 6)." has been replaced
by "We find that wider pegmatite dikes have a slower rate of Li loss (Figure 6)."

Line 297 "we find that" has been removed.

Line 301 "as an input parameter" has been added.

Line 303 "leading to" has been replaced by "resulting in".

Line 305 "There is an unexpected result:" has been replaced by "We found a surprising deviation
from the expected findings".

Line 339 "on account of" has been replaced by "because of".

Line 341 "whose" has been replaced by "and their".

Line 613 "think" has been replaced by "assume".

Line 616 "changing trend of" has been replaced by "changes in".

Line 618 "smaller than the" has been replaced by "within".

Line 650 "with" has been replaced by "that have"

Line 651 "Within the country rocks with different temperatures" has been removed.

Other punctuation errors and typing errors have also been corrected in the revised manuscript.

Response to Reviewer #2

Reviewer #2 (Remarks to the Author):

Review: Manuscript Number: NCOMMS-23-55961

Title: Pegmatite lithium deposits formed within low-temperature country rocks

The authors have carefully treated the suggestions, and the clarity and quality of the revised manuscript has been significantly improved. In this manuscript, a commonly overlooked, but potentially essential, factor in the genesis of pegmatite lithium deposits has been highlighted, which is a great contribution to this field of researches. With that being said, we still have some reservations regarding the robustness of the conclusion. Below, we give our major concerns that are based on geological considerations. We think that these concerns should be appropriately addressed before publication, and accordingly, a moderate revision is recommended.

We are very grateful to Reviewer #2 very much for providing insightful and helpful reviews, as well as sharing many excellent references! They greatly improved the manuscript. Thank you very much!

General comments:

In Fig. 3, two contrasting Li isotope and concentration trends are exhibited from pegmatite dikes outwards to wall rocks. For the trend around No. 134 pegmatite, the authors stressed that it is influenced by a nearby pegmatite dike. Please provide a photograph that clearly displays the spatial relationship between No. 134 and its nearby pegmatite dikes.

Thank you very much! The photograph showing the spatial relationship between No. 134 and its nearby pegmatite dikes is provided below.

In lines 208-220, the authors explained how the diffusion of Li can lead to the observed signatures of Li isotopes and contents of No. 433 pegmatite dike and its country rocks. However, this interpretation is quite confusing as the Li contents of surrounding wall rocks (894-1049 ppm) is distinctly higher than those of No. 433 dike (25.0-92.3 ppm). As diffusion is driven by differences

in element concentrations, it is hard to envisage that, if Li is diffused from pegmatite dike to wall rocks, the final Li content of wall rocks would be higher than the pegmatite dikes. Although the boundary layer theory was invoked in their discussion, we don't think the distinctly low Li contents in pegmatites can be explained as the result of the diffusion process.

Thank you very much! In the previous main text, we have provided multiple possible explanations for the data of No. 433 dike. Before providing the explanation mentioned above, we also emphasized that this is one of the possible explanations, as *“There may be multiple explanations for this pattern. For example, the No. 433 dike is originally Li-poor and has a very high $\delta^{7}\text{Li}$ value. Its country rocks may originally have a high Li content, or they were contaminated by other Li-rich pegmatites through diffusion. Here, we provide another possible scenario that...”* in the previous version of the manuscript.

This explanation contributes very little to the highlights and conclusions of our study. Although we have provided multiple possible explanations for the data of No. 433 dike, we tend to consider that the boundary layer effect affects the diffusion behavior of Li and may produce counterintuitive results in terms of Li contents in pegmatites and country rocks. However, discussing this topic needs more evidence and texts, and seems to be beyond the scope of this manuscript. Thus, we have deleted this explanation this time.

In line 230, the authors assume that the initial temperature of the pegmatite melt is 750°C, which is debatable. Several related information can be shown below: 1) It is estimated that Li-mineralized pegmatites have a mean crystallization temperature of -510 ± 25 °C compared to Be- and rare earth element (REE)-mineralized pegmatites with mean temperatures of -550 ± 45 °C (McCaffrey et al.,2023); 2) Granites, as the source of pegmatite, crystallized at about 700°C and could undergo overcooling to form pegmatites (London et al.,2018). So, the initial temperature of pegmatite melt should be modified.

London, D., 2018. Ore-forming processes within granitic pegmatites. *Ore Geology Reviews*, 101: 349-383.

McCaffrey, D.M. and Jowitt, S.M., 2023. The crystallization temperature of granitic pegmatites: The important relationship between undercooling and critical metal prospectivity. *Earth-Science Reviews*, 244: 104541.

Thank you very much! There may be a misunderstanding. The crystallization temperature of pegmatite (mostly between 400-600 °C) is definitely lower than the emplacement temperature of pegmatite (i.e., the initial temperature of the pegmatite melt). In the reference you recommended (McCaffrey and Jowitt, 2023), there is a text concerning this question, as *“Pegmatite magmas are emplaced at 700 ± 50 °C as evidenced by their near-haplogranite minimum compositions, lack of entrained crystals (London, 2014), and with emplacement temperatures measured from melt inclusions in pegmatite border zones (~ 720 °C; Thomas et al., 1988; Sirbescu et al., 2008).”* (page 4 in McCaffrey and Jowitt, 2023). Therefore, we think that assuming initial temperature of the pegmatite melt of 750 °C is reasonable. Moreover, in a recent thermal simulation by David London, the initial temperature of the pegmatite was also set to 750°C (page 365 in London, 2018).

In Fig.1c, the authors think that the age of granite intrusion and regional metamorphism are

consistent, which is debatable. Two main stages of regional metamorphism and deformation are documented in Jiajika. The regional metamorphism and granite intrusion are related to early-stage Barrovian metamorphism and late-stage Buchan metamorphism, respectively (Zhu et al., 2024). Here should be modified.

Zhu, J., Zhu, W., Xu, Z., Zheng, B., Li, G., Lin, H. and Gao, J., 2024. The composite metamorphic sequence in the Jiajika gneiss dome, Songpan-Ganze orogenic belt, eastern Tibet: P-T-D-t evolution and implications for lithium mineralization. GSA Bulletin. <https://doi.org/10.1130/B37747.1>

Thank you very much for sharing this excellent reference. In the revised manuscript, we have changed the descriptions, as:

“From the inner to outer aureoles, they are staurolite, staurolite-andalusite, andalusite, and biotite zones, respectively (Fig. 1b), which were formed by the superimposition of a Buchan-type metamorphism upon a Barrovian-type metamorphism (Zhu et al., 2024).”

“Country rocks near the plutons have high temperatures whereas distal rocks were at lower temperatures, as evidenced by the Buchan metamorphic series around the parent pluton in Jiajika (Zhu et al., 2024) as well as thermal metamorphism in other pegmatite districts⁶⁸.”

Specific comments:

In Fig.1, the “Majinzi” should be “Majingzi”;

Thanks very much! It has been modified in the revised manuscript.

In line 118, the “depoists” should be “deposits”.

Thanks very much! It has been corrected.